# Land-use emissions play a critical role in land-based mitigation for Paris climate targets

Anna B. Harper [1], Tom Powell [2], Peter M. Cox [1], Joanna House [3], Chris Huntingford [4], Timothy M. Lenton[2], Stephen Sitch[2], Eleanor Burke [5], Sarah E. Chadburn[1,6], William J. Collins [7], Edward Comyn-Platt [4], Vassilis Daioglou [8,9], Jonathan C. Doelman[8], Garry Hayman [4], Eddy Robertson[5], Detlef van Vuuren [8,9], Andy Wiltshire[5], Christopher P. Webber[7], Ana Bastos [10,11], Lena Boysen [12], Philippe Ciais [11], Narayanappa Devaraju[11], Atul K. Jain [13], Andreas Krause [14], Ben Poulter [15] & Shijie Shu [13]

Scenarios that limit global warming to below 2 °C by 2100 assume significant land-use change to support large-scale carbon dioxide ($CO_2$) removal from the atmosphere by afforestation/reforestation, avoided deforestation, and Biomass Energy with Carbon Capture and Storage (BECCS). The more ambitious mitigation scenarios require even greater land area for mitigation and/or earlier adoption of $CO_2$ removal strategies. Here we show that additional land-use change to meet a 1.5 °C climate change target could result in net losses of carbon from the land. The effectiveness of BECCS strongly depends on several assumptions related to the choice of biomass, the fate of initial above ground biomass, and the fossil-fuel emissions offset in the energy system. Depending on these factors, carbon removed from the atmosphere through BECCS could easily be offset by losses due to land-use change. If BECCS involves replacing high-carbon content ecosystems with crops, then forest-based mitigation could be more efficient for atmospheric $CO_2$ removal than BECCS.

[1] College of Engineering, Mathematics, and Physical Sciences, University of Exeter, Exeter EX4 4QF, UK. [2] College of Life and Environmental Sciences, University of Exeter, Exeter EX4 4QF, UK. [3] School of Geographical Sciences, University of Bristol, Bristol BS8 1SS, UK. [4] Centre for Ecology and Hydrology, Wallingford OX10 8BB, UK. [5] Met Office Hadley Centre, FitzRoy Road, Exeter EX1 3PB, UK. [6] University of Leeds, Leeds LS2 9JT, UK. [7] Department of Meteorology, University of Reading, Reading RG6 6BB, UK. [8] Department of Climate, Air and Energy, Netherlands Environmental Assessment Agency (PBL), PO Box 30314, 2500 GH The Hague, Netherlands. [9] Copernicus Institute of Sustainable Development, Utrecht University, Heidelberglaan 2, 3584 CS Utrecht, The Netherlands. [10] Department of Geography, Ludwig Maximilians University Munich, Luisenstr. 37, 80333 Munich, Germany. [11] Laboratoire des Sciences du Climat et de l'Environnement, LSCE/IPSL, CEA-CNRS-UVSQ, Université Paris-Saclay, 91191 Gif-sur-Yvette, France. [12] The Land in the Earth System, Max-Planck Institute for Meteorology, Bundesstrasse 53, 20146 Hamburg, Germany. [13] Department of Atmospheric Sciences, University of Illinois, Urbana, IL 61801, USA. [14] Karlsruhe Institute of Technology, Institute of Meteorology and Climate Research—Atmospheric Environmental Research (IMK-IFU), Kreuzeckbahnstr. 19, Garmisch-Partenkirchen 82467, Germany. [15] NASA GSFC, Biospheric Sciences Lab., Greenbelt, MD 20771, USA. Correspondence and requests for materials should be addressed to A.B.H. (email: a.harper@exeter.ac.uk)

The Paris Agreement set a target of "Holding the increase in the global average temperature to well below 2 °C above pre-industrial levels and to pursue efforts to limit the temperature increase to 1.5 °C above pre-industrial levels". However, equilibrium climate sensitivities projected by climate models suggest that current atmospheric greenhouse gas concentrations may already be very near those associated with a stable climate at 1.5 °C[1]. Hence, particularly strong near-term emissions reductions are needed in combination with greenhouse gas removal from the atmosphere (negative emissions) to achieve this lower temperature goal. Even achieving the 2 °C target will require transformational changes to energy provision and other sectors including industrial activity and land management, as well as negative emissions[2]. Governments have therefore asked the IPCC to assess the feasibility of stabilizing climate change at 1.5 °C, and the benefits of doing so.

Most of the scenarios considered in the IPCC 5th Assessment Report rely upon biomass energy with carbon capture and storage (BECCS) along with afforestation and reforestation to remove $CO_2$ from the atmosphere[3]. More recent studies also find a key role for land-based mitigation in contributing to a 2 °C target[4,5]. In the Integrated Assessment Model (IAM) scenarios consistent with a 2 °C target, a median of 3.3 GtC yr$^{-1}$ was removed from the atmosphere through BECCS by 2100, equivalent to one-third of present-day emissions from fossil fuel and industry. This median amount of BECCS would result in cumulative negative emissions of 166 GtC by 2100[6,7] and would supply ~170 EJ yr$^{-1}$ of primary energy. The bioenergy crops to deliver such a scale of $CO_2$ removal could occupy an estimated 380–700 Mha of land[7], equivalent to up to ~50% of the present-day cropland area[8]. There is high agreement from previous literature that 100 EJ yr$^{-1}$ of bioenergy could be produced sustainably, and moderate agreement that this can increase to 100–300 EJ yr$^{-1}$[9,10] (bioenergy currently supplies ~44.5 EJ yr$^{-1}$, but only 3% of that comes from dedicated bioenergy crops[7,10]). Scenarios targeting 1.5 °C tend to employ BECCS earlier than scenarios targeting 2 °C[11,12].

A second form of land-based climate mitigation is maintaining or growing forest carbon stocks. One study estimates that 1.1 GtCyr$^{-1}$ carbon dioxide removal is possible by 2100, requiring ~320 Mha of new forest[7]. A second study estimates greenhouse gas removal equivalent to 0.6–2.0 GtCyr$^{-1}$ based on potentials from afforestation/reforestation, avoided deforestation, natural forest management, forest plantations, fire management, and avoided woodfuel harvesting[5]. The $CO_2$ removal potential of forests also depends on the background climate and atmospheric $CO_2$ concentration[13].

To date there have been few studies with scenarios that directly evaluate land-based climate mitigation for a pathway targeting eventual temperature rise of only 1.5 °C, and those that do tend to focus on contributions from the energy[12,14] and agriculture[15–17] sectors. In addition, previous studies have shown a large uncertainty in carbon cycle responses to intensive land-based mitigation[18,19]. A key question is whether extensive land-based mitigation is likely to deliver the anticipated return in terms of carbon storage to achieve a 1.5 °C target.

Here, we explore the land-climate-carbon cycle interactions of a new scenario designed for 1.5 °C target temperature analyses produced by the IMAGE[20] IAM that includes afforestation/reforestation, avoided deforestation, and BECCS[21,22]. Bioenergy crops are modelled with the Joint UK Land Environment Simulator (JULES) under a range of climate and land-use change scenarios. We find that the simulated total land carbon storage is reduced with the land-use from the scenario designed for 1.5 °C climate change compared to the scenario designed for 2 °C, in contrast to the intended effect of the additional land-based mitigation in the 1.5 °C scenario. This is due to losses of vegetation and soil carbon when bioenergy crops replace high carbon ecosystems. Although JULES does not model high-yield bioenergy crops and does not account for impacts of bioenergy on reducing energy sector emissions, our results indicate that it is critically important to account for carbon-cycle impacts of replacing ecosystems with bioenergy crops.

## Results

**Land for food and bioenergy in the IMAGE scenario for 1.5 °C.** We analyze the impacts of the additional land-use change (LUC) to get from a 2 to a 1.5 °C world from a new scenario that leads to a forcing of 1.9 W m$^{-2}$ by 2100 compared to pre-industrial. The scenario is consistent with stabilization at or below 1.5 °C and was produced by IMAGE using a mitigation pathway with moderate challenges for adaptation and mitigation[15] (Shared Socio-economic Pathway 2, SSP2-RCP1.9 or IM1.9). Land-based mitigation options are part of the overall mitigation portfolio, while maintaining an assumption that food production for the global population is the dominant driver of global land use. There are a number of important assumptions associated with the IMAGE scenario, including the use of high-yield bioenergy crops (the majority of which are *Miscanthus*, with some coppiced tree plantations), replacement of fossil-fuel based power generation in the energy system, the use of agricultural and forestry residues as a biomass energy resource, and effective storage of the captured carbon. IMAGE also assumes that in the case of direct deforestation for bioenergy crops, nearly all of the original above-ground biomass is put into long-term carbon pools, equivalent to assuming it is used for BECCS. In IM1.9, land for bioenergy crops rapidly expands from 2030 to 2050, reaching a maximum of 550 Mha by 2060, and declining to 430 Mha by 2100 (Fig. 1). In comparison, a scenario that is more likely to stabilize close to 2 °C by 2100 (SSP2-RCP2.6, or IM2.6) allocates a lower maximum of 325 Mha to bioenergy crops in 2085. Approximately 60–70% of the bioenergy crops are used with CCS in IMAGE, and agricultural and forestry residues make up at least 40% (70%) of the bioenergy feedstock in IM1.9 (IM2.6). Based on these numbers and simulated biomass yields in IMAGE, a total of 130 GtC (20 GtC) is stored in geologic reservoirs via BECCS from dedicated bioenergy crops in the IM1.9 (IM2.6) scenario by 2100.

**Earth system impacts of land-based mitigation.** In IMAGE, carbon cycle impacts of BECCS are evaluated using the dynamic global vegetation model (DGVM) LPJml. In this study, bioenergy crops are specified from the land-use maps from the IMAGE IM1.9 and IM2.6 scenarios (Table 1). Yields are simulated by the JULES[23] DGVM based on harvesting of natural C3 and C4 grasses. We then calculate the potential carbon stored via BECCS based on permanently storing 60% of carbon from harvested biomass (compared to 50–52% from previous studies[18,24,25] and to 77–87% in IMAGE). JULES is driven by regional and seasonal climate change patterns from 34 CMIP5[26] Earth System Models (ESMs) based on the IMOGEN pattern-scaling method[27]. IMOGEN is a simplified coupled carbon-climate model without biophysical feedbacks (Methods). We force these patterns with prescribed global temperature time series that approach 1.5° and 2 °C warming targets by 2100 (Methods)[28]. The use of JULES-IMOGEN allows climate change impacts to be included in an assessment of the effectiveness of and possible risks or benefits of large scale land-use change for climate mitigation as assumed in IAM scenarios. The JULES carbon fluxes and carbon stocks have been validated against available observations[23,29,30], suggesting that the modelled carbon turnover times are realistic in most ecosystems (see Methods and Table 2).

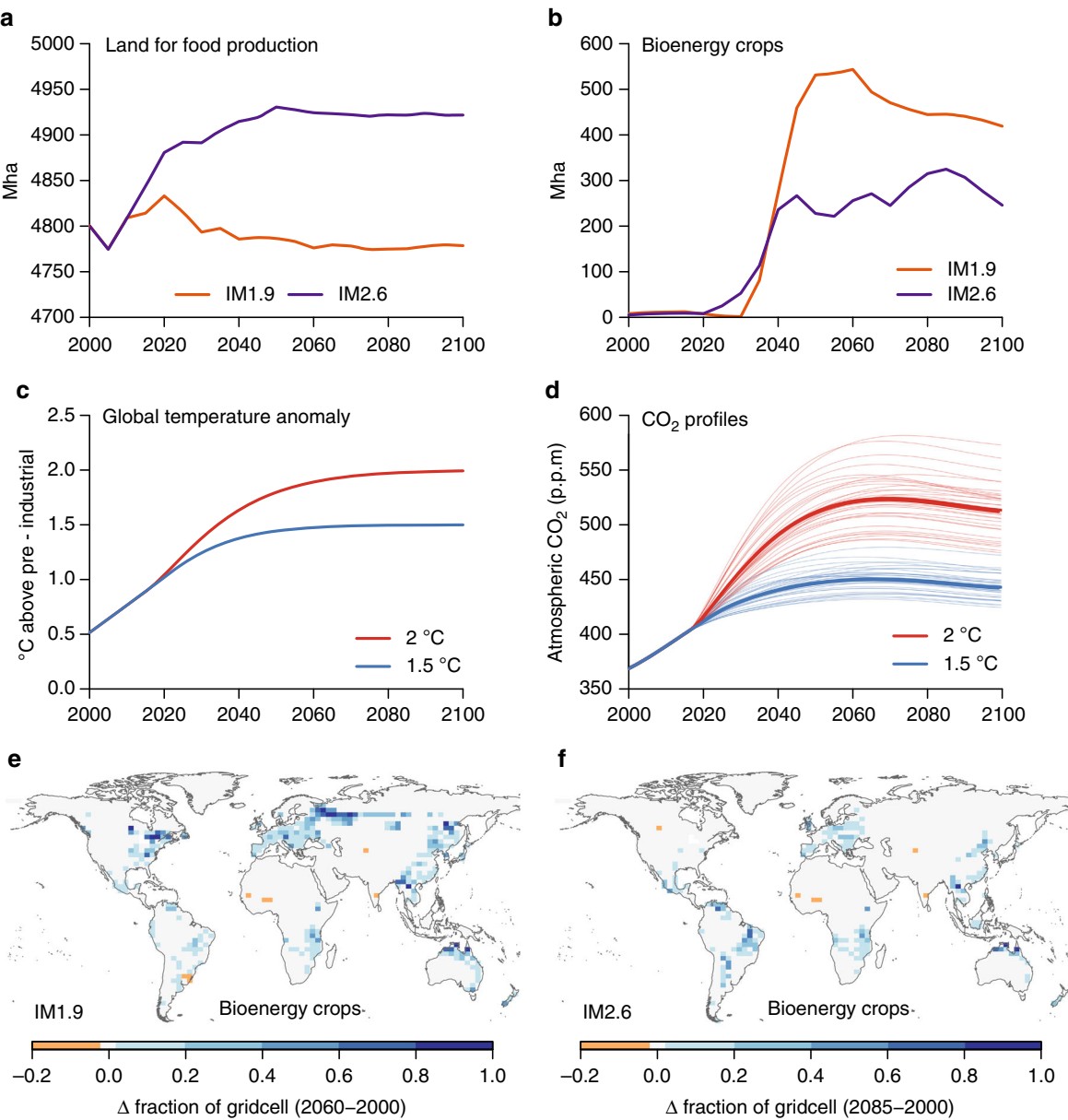

**Fig. 1** Scenarios for land-use and climate change. **a**, **b** Land used for food production (crops and pasture) and bioenergy crops from the IMAGE SSP2 scenarios IM1.9 and IM2.6 (available from https://data.knmi.nl/datasets?q=PBL). **c** Temperature profiles for the idealized scenarios reaching nearly 1.5 °C and 2 °C by 2100. **d** $CO_2$ concentrations for each of the 34 ESMs emulated with IMOGEN. The $CO_2$ concentrations relate to the temperatures in **c** depending on each model's climate sensitivity (Methods). **e**, **f** Spatial maps of change in land for bioenergy crops in IM1.9 and IM2.6. For each scenario, the change is shown as the difference between 2000 and the year of maximum extent of bioenergy crops (2060 for IM1.9 and 2085 for IM2.6)

**Table 1 Summary of experiments. IM1.9 and IM2.6 refer to the IMAGE versions of the Representative Concentration Pathways for 1.9 and 2.6 W m⁻² radiative forcing by 2100 (see Fig. 1e, f for a snapshot of changes in each scenario)**

| Name of experiment | Land-use change | Climate change by 2100 | Transient atmospheric $CO_2$ |
|---|---|---|---|
| 1.5 °C_IM19 | IM1.9 | 1.5 °C | Diagnosed based on temperature profile |
| 1.5 °C_IM26 | IM2.6 | | |
| 2 °C_IM19 | IM1.9 | 2 °C | |
| 2 °C_IM26 | IM2.6 | | |
| 2 °C_IM26_1.5CO2 | IM2.6 | | Diagnosed based on 1.5 °C temperature profile |

Temperature change by 2100 is prescribed using idealized changes that asymptote to either 1.5° or 2 °C. The atmospheric $CO_2$ concentration is diagnosed based on the associated temperature profile and each climate model's climate sensitivity (Methods). The 2 °C_IM26_1.5CO2 experiment has a 2 °C climate change profile but the $CO_2$ concentrations from the 1.5 °C temperature profile

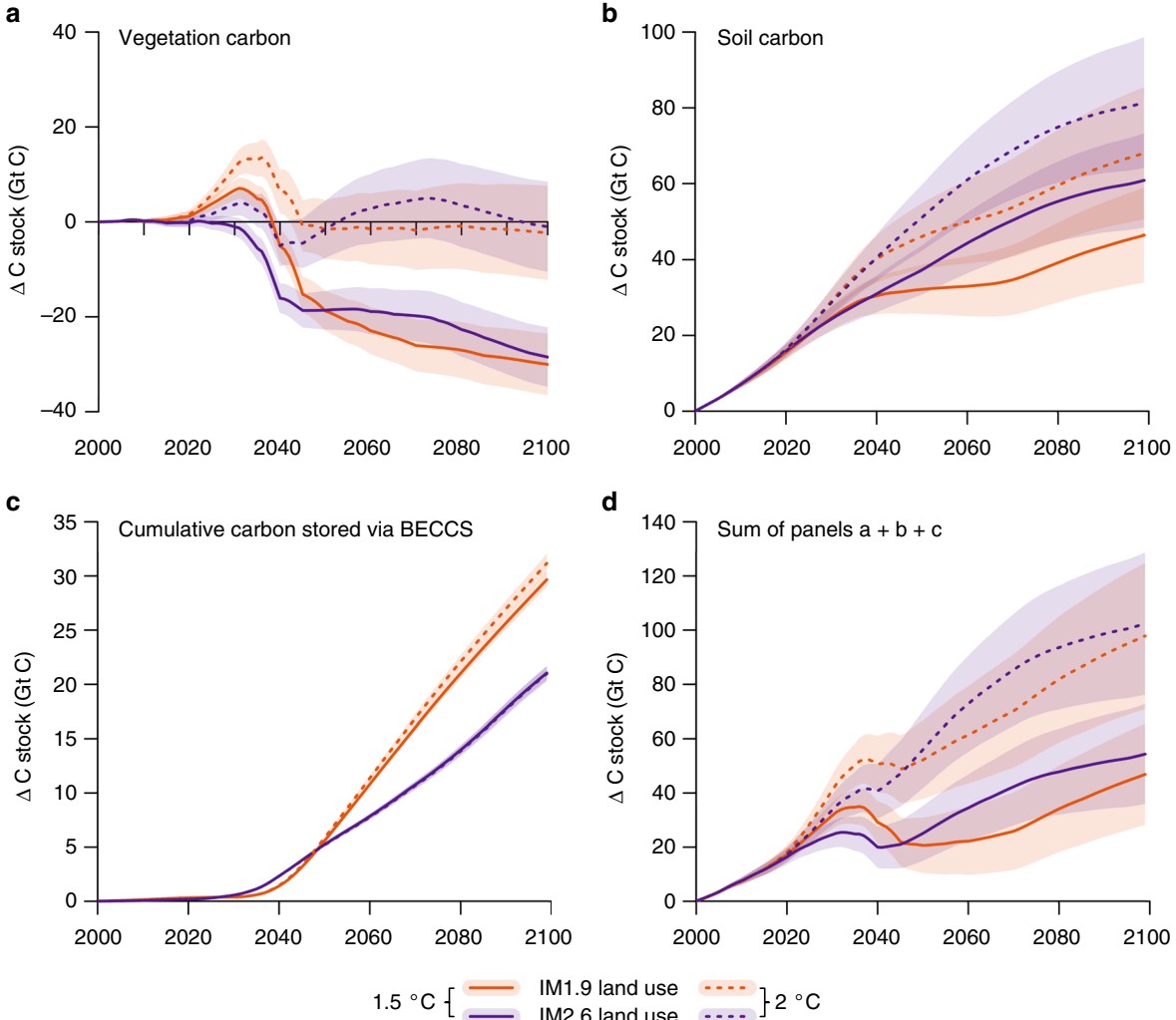

**Fig. 2** Carbon cycle responses to two land-use and two climate stabilization scenarios simulated by the land surface model JULES. Land-use scenarios are shown with different colours: IM1.9 (orange) and IM2.6 (purple), and the climate scenarios are shown with different line patterns: 1.5 °C (solid) and 2 °C (dashed) above preindustrial by 2100. Panels show simulated vegetation (**a**) and soil carbon (**b**), the cumulative storage of carbon through BECCS (**c**), and the total land carbon stock (including captured carbon via BECCS) (**d**). Shading shows ± 1 standard deviation from the ensemble mean from the 34 ESM climates represented in IMOGEN

Under the modelled land-use and climate scenarios we find that the accumulated carbon removed from the atmosphere through BECCS is largely offset by initial reductions in stored land carbon. Our results suggest a land carbon sink that is twice as strong in the 2 °C scenario compared to 1.5 °C (Fig. 2), irrespective of land-use scenario. This is due to both the fertilizing effect of $CO_2$ being larger, and the growth of more high latitude vegetation in the 2 °C scenario. These positive impacts on land carbon of the 2 °C scenario are partially offset by losses of carbon due to higher respiration rates at 2 °C compared to 1.5 °C. We discuss these findings in more detail below.

The total land carbon storage ($C_{veg} + C_{soil}$ + geological storage from BECCS) is lowest in the 1.5 °C_IM1.9 scenario, which is in direct contrast to the intended effect of the additional land-based mitigation in this scenario. The net change in carbon storage in 1.5 °C_IM1.9 from 2000 to 2100 is +47 ± 18 GtC, compared to + 102 ± 25 GtC in 2 °C_IM2.6 (Fig. 2d) (unless otherwise specified, reported numbers are the ensemble mean across 34 GCMs simulated in JULES-IMOGEN ± 1 standard deviation). There is a loss of 30 ± 6 GtC vegetation carbon, $C_{veg}$, in the 1.5 °C_IM1.9 scenario compared to almost no change (−1 ± 9 GtC) in the 2 °C_IM2.6 scenario. Soil carbon, $C_{soil}$, increases

for all scenarios, particularly in the high latitudes (Fig. 3), where there is an increase in woody vegetation (Fig. 4). The increase in $C_{soil}$ is greatest for 2 °C_IM2.6 ( + 81 ± 17 GtC compared to + 46 ± 12 GtC for 1.5 °C_IM1.9). As discussed below, a large portion of the benefit in the 2 °C scenario for $C_{veg}$ and $C_{soil}$ is due to $CO_2$ fertilization. The geological carbon storage via BECCS is 30 ± 1 GtC by 2100 for 1.5 °C_IM1.9 and 21 ± 1 GtC by 2100 for 2 °C_IM2.6, compared to 130 GtC and 20 GtC in IMAGE. Critically, we find that for the IM1.9 scenario, the JULES BECCS storage is < 25% that simulated by IMAGE, despite JULES using the harvest from all bioenergy crops with CCS, while in IMAGE at most 60–70% of bioenergy crops are used with CCS.

The relative impacts of the climate and land-use scenarios are also apparent in Fig. 2, showing a greater effect of climate scenario on vegetation carbon and mixed results for soil carbon. We estimate the climate effect as the difference in ΔC (the change in carbon from 2000 to 2100) between two climate scenarios with the same LUC (IM1.9) but different climates. Similarly, the LUC effect is estimated as the difference in ΔC between two scenarios with the same climate (1.5 °C) but different LUC. The climate effect on $C_{veg}$ results in higher $ΔC_{veg}$ by 28 GtC in the 2 °C climate, while the LUC effect results in only 1 GtC more in IM2.6

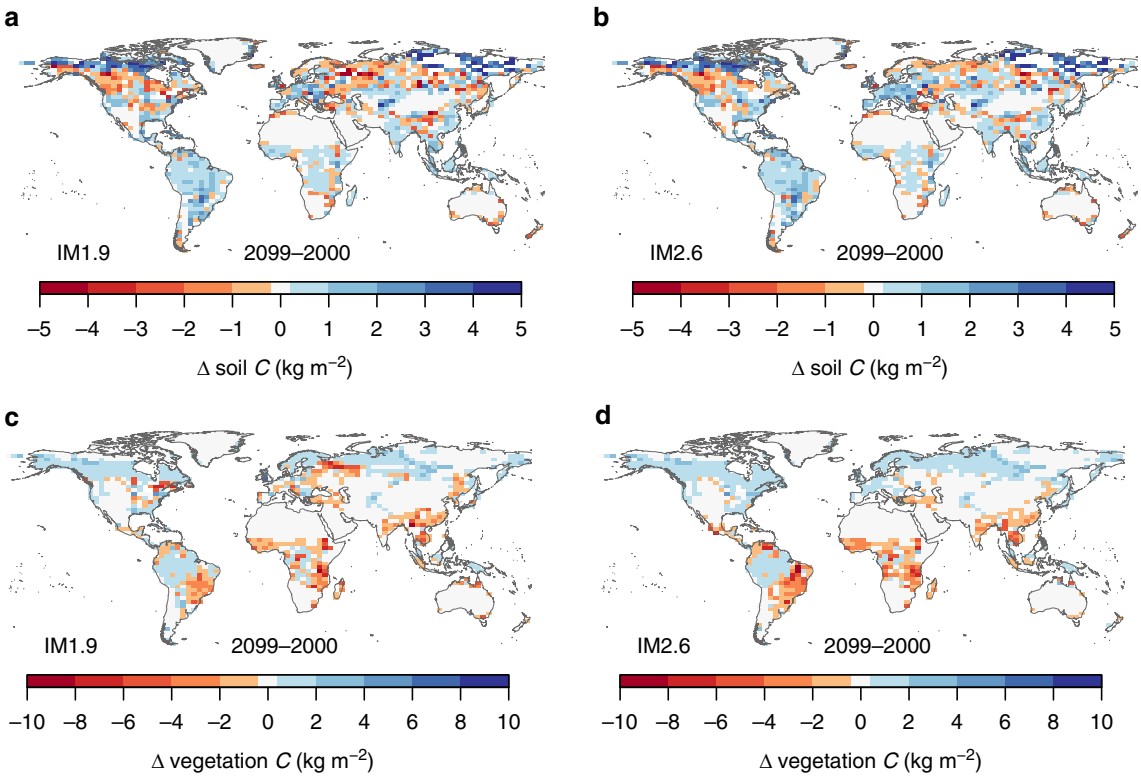

**Fig. 3** Simulated changes in soil and vegetation carbon due to land use and climate change. Carbon stored in soils (**a**, **b**) and vegetation (**c**, **d**) is simulated by the land surface model JULES. The changes are shown in kg m$^{-2}$ from 2000 to 2099 for the IM1.9 (**a**, **c**) and IM2.6 (**b**, **d**) land-use scenarios with the 1.5 °C climate change scenario

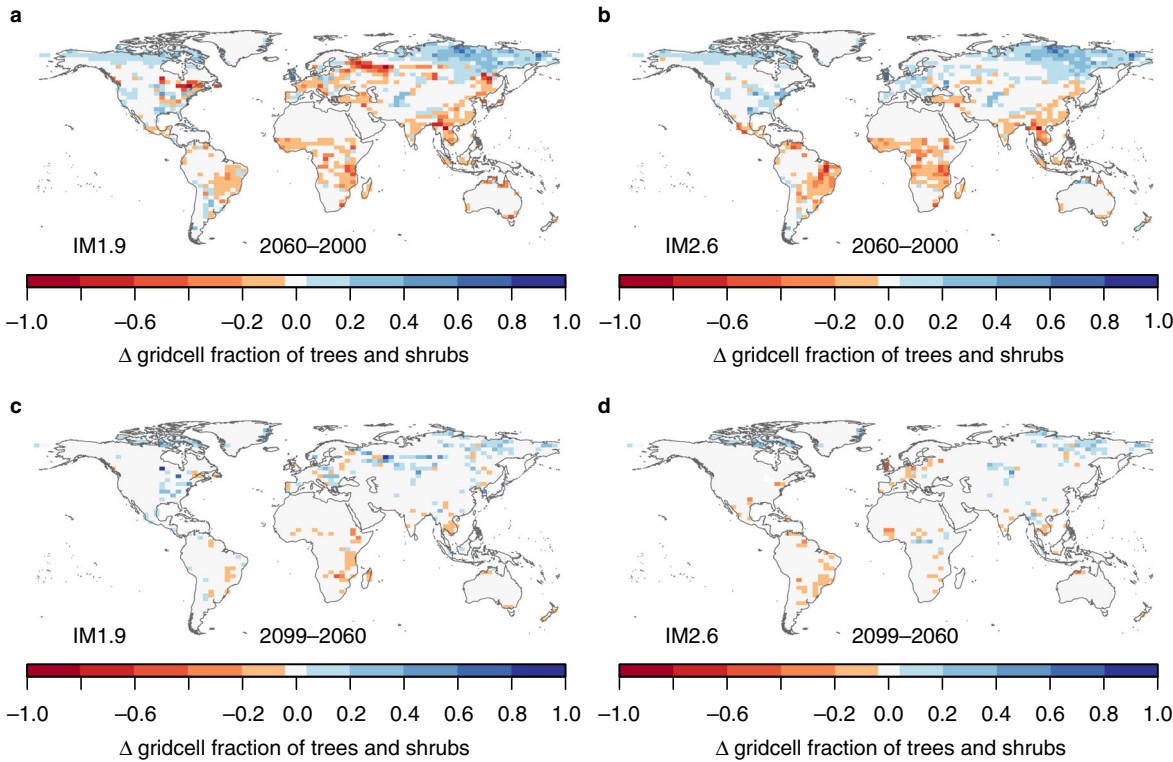

**Fig. 4** Changes in tree and shrub coverage simulated by the land surface model JULES (units = fraction of grid cell). The changes are from 2000 to 2060 (**a**, **b**) and from 2060 to 2099 (**c**, **d**) for the two land-use scenarios with the 1.5 °C climate change scenario

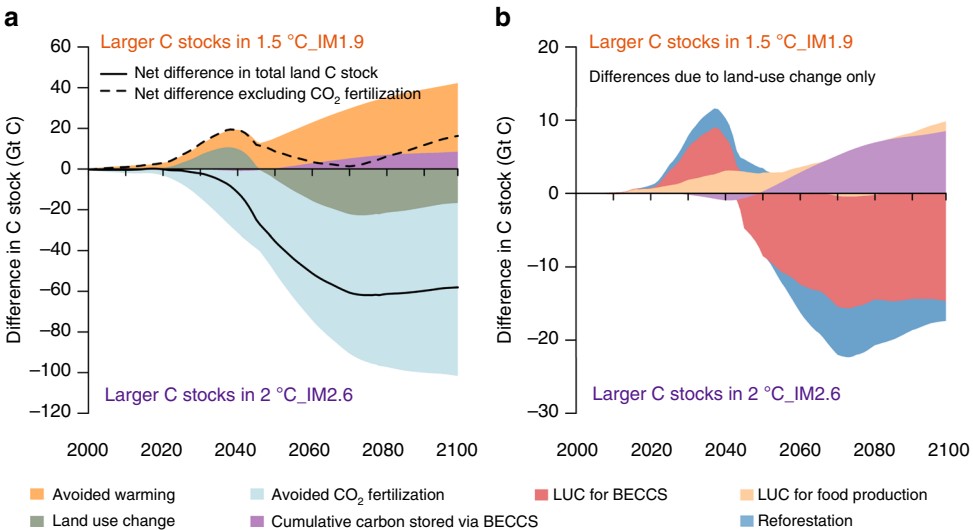

**Fig. 5** Attribution of differences in land carbon stocks between scenarios based on JULES model simulations. **a** The net difference (sum of $C_{veg} + C_{soil}$ + geologically stored $CO_2$ from BECCS) and each component between 1.5 °C_IM1.9 and 2 °C_IM2.6. Positive values indicate larger land carbon stocks in 1.5 °C_IM1.9. **b** The drivers of the land use change effect. Total LUC effect (olive green in panel **a**) can be attributed to LUC for: food production (crop and pasture), BECCS, and land abandonment/reforestation. These are shown with the cumulative carbon removed via BECCS for comparison

(similar effects are found when estimating the climate effect with IM2.6 and the LUC effect with 2 °C warming). The largest increases in vegetation carbon due to climate change occur in boreal and extratropical forests (Fig. 3). This result indicates strong capacity for undisturbed forests to continue as a carbon sink in the future. Any deforestation results in both immediate losses of carbon and a lost sink capacity.

There is substantial overlap between the impacts of the climate and LUC scenarios on $C_{soil}$, with a ~50% greater effect from climate compared to land-use. The climate effect results in 20 GtC more $C_{soil}$ in the 2 °C climate, and the LUC effect results in 13 GtC more $C_{soil}$ in the IM2.6 scenario. Climate change has a substantial impact on high latitude ecosystems: soil carbon increases by 21–26 GtC over the 21st century in the tundra biome in both IM1.9 scenarios. We compare this increase to an additional experiment with climate change from the 2 °C scenario but $CO_2$ from the 1.5 °C scenario (2 °C_IM2.6_1.5CO2; land-use change in the tundra is negligible). $C_{soil}$ increases by 21 GtC in 1.5 °C_IM1.9 and by 22 GtC in 2 °C_IM2.6_1.5CO2, indicating that the increase in this region is mostly due to warmer temperatures (rather than $CO_2$ fertilization) that encourage expansion of needle-leaf trees and shrubs (Fig. 4), and counteract effects of higher soil respiration. Overall, LUC for mitigation in IM1.9 has a small net negative effect on the land carbon, while the climate change between 1.5 °C and 2 °C has a larger positive effect.

The net difference in total land carbon storage between the 1.5 °C_IM1.9 and 2 °C_IM2.6 scenarios is −58 GtC (Fig. 5a) (here negative values indicate relatively more carbon stored in the 2 °C_IM2.6 scenario). However, to achieve the ambitious 1.5 °C climate change target, more land carbon storage would be required in the 1.5 °C_IM1.9 scenario. The attribution of changes in total land carbon is shown in Fig. 5a (Methods). Warmer temperatures tend to reduce land carbon, mostly due to increased heterotrophic respiration, so the reduced climate change in the 1.5 °C scenario maintains more carbon on the land ( + 34 GtC). Almost half of this benefit is negated, however, by land-use changes (−17 GtC). Thus, the land-use change for 1.5 °C releases ~50% of carbon kept in the land reservoirs through avoided warming. The largest and most uncertain term in the attribution is the $CO_2$ fertilization effect (−85 GtC). We remove the differential impact of $CO_2$ fertilization by comparing the

2 °C_IM2.6_1.5CO2 scenario to the 1.5 °C_IM1.9 scenario. The latter has a net benefit of 16 GtC compared to 2 °C_IM2.6_1.5CO2, due to avoided warming and a larger BECCS flux based on the IM1.9 land-use.

To determine the impacts of separate land-use change drivers on stored land carbon, we categorize the changes based on the land-use change occurring in each grid cell for each year: LUC for food, bioenergy crops, or afforestation/reforestation (Methods). The 1.5 °C_IM1.9 scenario begins the century with relatively higher land carbon due to lower increases in land-use for food and bioenergy compared to 2 °C_IM2.6, especially in the tropics (Fig. 1). During the period 2035–2045, land-use for bioenergy crops increases rapidly in IM1.9 and there are large losses of vegetation and soil carbon in eastern North American and northern Eurasia, leading to less land carbon in these regions with 1.5 °C_IM1.9 (Fig. 6a, b). Many of the land-use changes are short-lived, and after 2060 many high latitude bioenergy plantations are abandoned in IM1.9. JULES simulates regrowth of trees and shrubs in these regions (Fig. 4), so vegetation carbon increases, but because the younger, smaller trees have lower carbon inputs, loss of soil carbon continues during this regrowth phase (not shown).

In order to check for model-dependency in our results, we compare the 1.5 °C _IM1.9 and 1.5 °C _IM2.6 scenarios in five other DGVMs driven by a subset of the climates from the JULES-IMOGEN simulations (Methods). All models simulate higher increases in land carbon from 2015 to 2100 in IM2.6 compared to IM1.9. The average difference in land carbon storage between scenarios from the five models is 23 GtC (not including JULES), with a minimum of 2 GtC and a maximum of 40 GtC, compared to 9 GtC in JULES. On average, $C_{veg}$ is 18 GtC higher (range of 2–40 GtC) and $C_{soil}$ is 4 GtC higher (range of −6.7 to 16 GtC) with the IM2.6 land-use. For the same subset of climates, JULES simulates higher $C_{veg}$ and $C_{soil}$ of 2 GtC and 15 GtC, respectively, with the IM2.6 land use.

**Comparison of forests and BECCS for climate mitigation.** Soil carbon plays an important role in determining the effectiveness of BECCS for climate mitigation. The continued emissions of soil carbon following land-use change result in significant payback

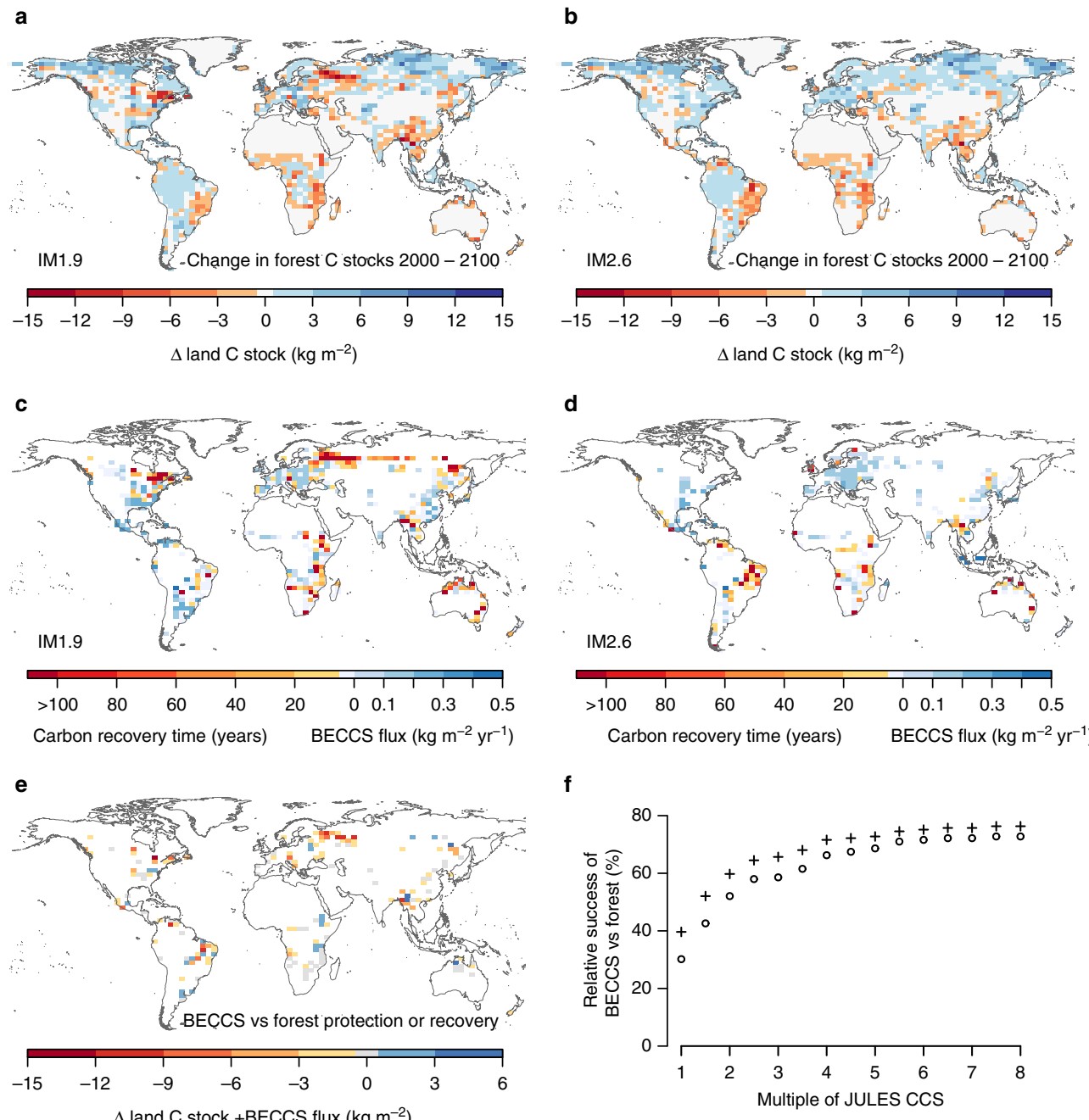

**Fig. 6** Comparison of forests and bioenergy crops with carbon capture and storage for carbon storage based on JULES simulations. **a**, **b** Change in forest carbon stocks over the 21st century (vegetation, soils and woody product pools) for the 1.5 °C climate scenario with IM1.9 and IM2.6 land use patterns. **c**, **d** Recovery time for BECCS. Blues indicate the recovery time is 0 years, and instead shows the mean annual flux of captured carbon via BECCS. **e** Difference in total carbon stocks (including accumulated storage via BECCS) at 2100 on grid cells where the two scenarios have conflicting land-use change. The convention is: scenario with bioenergy crops minus scenario with forests, such that blues indicate more carbon stored with BECCS and reds indicate more carbon stored with forests. **f** The percentage of points in **e**, in which BECCS is more successful at accumulating carbon than forest preservation or reforestation, showing the effect of increasing the default carbon storage from BECCS in JULES. Crosses indicate the benefit of harvesting initial aboveground biomass for BECCS as in IMAGE

times in many places where bioenergy crops are planted in IM1.9. We quantify the payback time (years), or recovery time, as the change in carbon stocks attributed to LUC for bioenergy crops (Methods) divided by the average annual flux of carbon into the BECCS geologic reservoir (Fig. 6c, d). JULES shows that recovery time is insignificant when bioenergy crops replace existing agriculture, for example in Europe and eastern North America in 2 °C_IM2.6. In the tropics, recovery times can be 10–100 + years

due to either very low productivity or very high pre-existing soil carbon content. The significant losses of soil carbon at high latitudes result in recovery times of >100 years.

We examine the relative success of BECCS compared to forest-based mitigation by comparing the change in carbon content (including captured carbon via BECCS, and excluding changes in land carbon due to food production) in grid cells where one scenario has bioenergy crops while the other scenario retains or

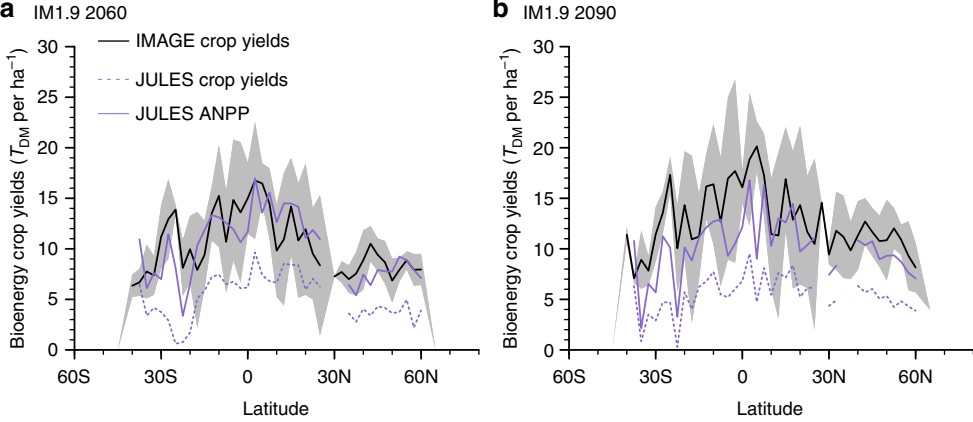

**Fig. 7** Comparison of bioenergy crop yields as simulated by JULES and IMAGE. Zonal mean non-woody potential biomass from IMAGE (black lines) ± 1 standard deviation (shading) in 2060 (**a**) and 2090 (**b**) with IM1.9 compared to JULES: solid line is the aboveground NPP, and the dashed line is the harvest on bioenergy crop tiles

grows forests (Fig. 6e, f). This comparison is based purely on changes to the land carbon budget and does not account for fossil fuel offsets due to using bioenergy crops. The representation of BECCS is more successful in only 27% of the grid cells where land-use differs between scenarios under our default JULES configuration. However, these estimates represent a lower boundary for potential $CO_2$ removal via BECCS. Doubling the predicted carbon storage from BECCS in JULES (which could be achieved through higher yields, including residues in feedstocks, or more efficient processing, transport, capture and storage) increases the fraction of grid cells where BECCS is a good option to 48%, although globally the land carbon stocks in IM1.9 would begin to match those in IM2.6 (Fig. 2c, d). Even if we assume highly productive bioenergy crops and efficient BECCS, in JULES > 20% of BECCS grid cells fail to break even with regard to carbon stored during the 21st century. These are grid cells with high payback times discussed above. BECCS appears more favorable if the original vegetation carbon stock prior to LUC for bioenergy crops is also captured via BECCS, as is assumed by IMAGE (black crosses in Fig. 6f), instead of being stored in product pools that gradually decay as assumed by JULES (Methods). Globally, forests store 861 ± 66 GtC, with 44% in the top 1 m of soils[31]. Removal of a forest results in both the immediate loss of above-ground carbon, and losses in soil carbon as a result of reduced litterfall inputs. The loss of vegetation carbon is a one-time loss that is avoided in IMAGE due to the assumption that initial harvested aboveground biomass from forests contributes to bioenergy. However, the loss of soil carbon in regions with high carbon density makes it difficult for BECCS to result in a net negative emission of $CO_2$.

## Discussion

The relative effectiveness of BECCS compared to afforestation/reforestation with avoided deforestation depends on several factors including: previous land cover, the initial loss/gain of carbon due to land-use change, transient changes in soil carbon, the yield of bioenergy crops, the amount of harvested carbon that is ultimately stored underground, and the permanence of regrowing/maintained forests. IMAGE assumes higher yields based on purpose-based bioenergy crops modelled by LPJml, the use of initial above ground biomass harvested in boreal forests for BECCS, and replacement of fossil-fuel based emissions in the energy system, which all make BECCS a viable option for climate mitigation. The replacement of fossil-fuel emissions provides a significant advantage for the use of BECCS in IMAGE, which

explains its adoption in IAM projections. For the IM1.9 scenario, BECCS stores ~4.3 times more carbon in IMAGE compared to JULES (130 GtC compared to 30 GtC by 2100). This is a result of differences in crop yields, conversion efficiency, or assumptions about the use of residues or proportion of bioenergy crops used with CCS. We expand upon these differences below.

Based on land-use in IM1.9 in 2100, the average aboveground net primary productivity (ANPP) from the bioenergy crops in JULES is 10.4 tonnes of dry mass ($T_{DM}$) ha$^{-1}$ y$^{-1}$ (for the 1.5 °C scenario), compared to an average yield from non-woody bioenergy crops of 15.8 $T_{DM}$ha$^{-1}$ y$^{-1}$ in IMAGE (Methods). Zonal mean ANPP from JULES (2–17 $T_{DM}$ha$^{-1}$ y$^{-1}$) is comparable to the IMAGE harvested biomass (4–20 $T_{DM}$ ha$^{-1}$ y$^{-1}$) (Fig. 7). However, JULES calculates harvest as 30% of litterfall, which results in bioenergy crop yields that are ~50% of those predicted in IMAGE.

After harvest, biomass is transported from fields to power stations, processed, and combusted. The captured carbon is then transported to the geological storage site and stored. In JULES, an efficiency factor of 0.6 is applied based on losses that would occur during the journey from harvest to final storage. In total 50% efficiency was assumed in ref. [18], which also represented all carbon losses from harvest to storage. In comparison, a case study of switchgrass used for bioenergy at a co-firing facility with CCS found an accumulated loss of 52% of the harvested biomass[24,25]. IMAGE assumes that 10% of harvested biomass is lost prior to use for BECCS. Most CCS technology can capture up to 85–97% of the carbon at the plants (IPCC 2005), resulting in a net efficiency in IMAGE of 77–87%. Therefore the JULES efficiency results in a final storage of C that is less optimistic than IMAGE (60% vs up to 87% efficiency). Emissions along the way are accounted for in the transportation and energy systems[32] in IMAGE, but are included in the JULES efficiency factor.

Combining the above differences between JULES and IMAGE explains a portion of the gap in predicted carbon captured via BECCS. As an illustration, assuming an equal area of bioenergy cropland, crops in IMAGE could produce 1 GtC initial biomass, whereas JULES productivity would result in 0.5 GtC. Assuming 60–70% of the bioenergy crops in IMAGE are used with CCS with an 87% efficiency results in 0.52–0.61 GtC captured in the IMAGE system. For JULES, all of the harvest goes to a CCS facility, and this process has a 60% efficiency, resulting in 0.3 GtC captured. In this comparison the carbon stored via BECCS from the same area of cropland is ~1.5–2 times higher in IMAGE than in JULES. We also evaluate the carbon captured from BECCS in LPJ-GUESS, the only other DGVM in this analysis that includes a

representation of BECCS, in this case based on maize[33]. By 2100 and assuming the same transfer efficiencies and emissions as in JULES, the carbon captured via BECCS in LPJ-GUESS is 73 GtC in IM1.9 and 57 GtC in IM2.6. These numbers are 2.5–3 times higher than predicted by JULES (and also assume that all bioenergy crops are used with CCS). These comparisons indicate that most likely the carbon captured with BECCS in JULES should be increased by a factor of 1.5–3 (Fig. 6f). Critically, this would still result in ~35–40% of the grid cells with a higher $CO_2$ removal potential from afforestation, reforestation, and avoided deforestation than from BECCS, although the study does not account for the avoided emissions from the electricity produced.

It is important to note that this study only considers carbon cycle implications of land use change. Other important factors that would affect climate change include emissions of other greenhouse gases and biophysical changes to the land surface. In addition, land-use change alters several ecosystem services, and large-scale bioenergy croplands for BECCS could put additional pressure on freshwater systems, food security, land use and biodiversity, and biogeochemical cycles[18,33,34].

Agricultural practices can result in emissions of both $N_2O$ and $CH_4$. JULES calculates methane emissions from wetlands and permafrost thaw, the former in proportion to the simulated wetland area[35]. Land-use change has small effects on wetland methane emissions (< 2% difference simulated by JULES). A more substantial factor, although not considered in this study, could be the 100 Mha reduction in pastureland by 2100 in IM1.9 compared to IM2.6. Methane mitigation is critical for limiting climate change, and has the added benefit of reducing surface ozone, which benefits humans and natural ecosystems, allowing for more productive crops and increased uptake of $CO_2$ by vegetation[36].

We assumed modelled crops to be non-fertilized and rain-fed, but fertilizer inputs would result in emissions of $N_2O$. In IMAGE, it is assumed that bioenergy crops only receive low amounts of fertilizer. The total assumed fertilizer inputs are similar between scenarios (there is about 0.1 more Gg fertilizer applied in RCP1.9 between 2015 and 2100). Similar to JULES, IMAGE assumes that bioenergy crops are rain-fed only. We find regional changes in JULES predicted runoff due to the land-use change: there is decreased runoff between the present-day and future in IM1.9 in northeastern US/eastern Canada and in the boreal forest region of western Russia. These regions coincide with large-scale deforestation for bioenergy crops, and as expected the loss of deep-rooted trees reduces the evaporation and increases runoff. In the tropics, there are some reductions in runoff in IM1.9 compared to IM2.6, as these are regions with increased tree cover in the former. These changes would have impacts on water resources[33] and hydroelectric power generation.

Land-use change also affects the climate by altering biophysical properties of the land (for example: albedo, sensible and latent heat flux, roughness length)[37]. Forests can have a warming effect by absorbing more solar radiation, and a cooling effect due to deeper rooting systems than crops and higher rates of evapotranspiration (ET). The net biophysical effect depends on location: the albedo effect tends to dominate at high latitudes, while the ET effect tends to dominate in the Tropics. A reversal of past deforestation should result in a net cooling effect[13,38]. When bioenergy crops are placed on pre-existing agricultural land, the biophysical impacts are small. It is possible that the biophysical impacts of IM1.9 would be a net cooling due to increased tropical forests and decreased boreal forests. Land-use change also impacts extreme weather events such as daytime high temperatures[39,40]. Many of these effects require evaluation in a coupled GCM framework, to fully capture local and regional land-atmosphere feedbacks.

Despite the challenges, BECCS remains a potential method for climate mitigation, particularly due to its relatively low expense compared to other forms of $CO_2$ removal and because it is an energy provider[7,41]. In addition, sustainable levels of bioenergy can lead to co-benefits such as income generation, energy independence, improved water use efficiency, increased biodiversity, and soil carbon retention (although potential for negative impacts are also possible and examples of good practice should be followed[9]). Additional barriers to BECCS include the lack of current infrastructure for large-scale power generation with biomass and subsequent CCS, the need for governance and monitoring of CCS facilities, and public perception of geologic storage of $CO_2$[42]. Storage reservoirs without leaks are essential to make any investment in BECCS worthwhile from a carbon capture perspective[24].

In conclusion, we have shown that different assumptions and modelling frameworks between dynamic global vegetation models and integrated assessment models can give very different conclusions regarding the effectiveness of land-based climate mitigation techniques. The carbon cycle implications of these mitigation methods need to be accounted for. In JULES, different assumptions show that additional reliance on BECCS in regions where bioenergy crops replace ecosystems with high carbon contents could easily result in negative carbon balance and therefore may be unwise. In these cases, forest conservation and afforestation/reforestation are more effective methods to increase the budget of carbon emissions for stabilization at 1.5 °C. In the case of replacing carbon-dense ecosystems with bioenergy crops, consideration should include the lost potential for $CO_2$ uptake by forests under even moderate levels of climate change. Forest-based mitigation also has a wide range of co-benefits for ecosystems and humans, including biodiversity, income generation, flood control, and improving soil, air and water quality[5]. Although BECCS could be beneficial with adequate sustainability constraints applied[43–45], caution is needed in determining land-use change fluxes.

## Methods

**Modelling the terrestrial carbon cycle.** JULES (the Joint UK Land Environment Simulator) is a land surface model that calculates the turbulent exchange of $CO_2$, heat, water, and momentum between the land and atmosphere[46,47]. JULES calculates net primary productivity (NPP) on a half-hour time-step and updates carbon stores and plant functional type (PFT) areas every ten days. The dynamic global vegetation model TRIFFID updates the vegetation and soil carbon pools, and simulates competition for space between PFTs[47,48].

Nine natural PFTs[30] are simulated, along with two crop PFTs and two pasture PFTs. The natural, crop, and pasture PFTs do not compete with each other; they grow within prescribed natural, cropland and pasture areas[49]. However, crop and pasture PFTs have the same parameters as natural C3 and C4 grasses[30]. TRIFFID ensures a carbon balance, so all NPP is accounted for either through increased carbon density, increased PFT coverage or litter. After the new PFT fractions areas are calculated, litter ($\Lambda$) is calculated based on the change to vegetation area and carbon density:

$$\Lambda = \Pi\nu_{n-1} - \left( C_{veg}\nu - C_{veg,n-1}\nu_{n-1} \right)$$

Where is NPP accumulated over the 10-day period, $\nu$ is the fraction of the PFT in the grid cell and the subscript $n$-1 refers to the value at the previous TRIFFID time step.

When land is first cleared for agriculture, woody biomass is put into three woody product pools to represent the products of wood harvesting[50]. Below-ground carbon goes directly into the fast-decay pools in the soil, while the above-ground biomass is allocated depending on PFT: for trees 10% of the litter goes into a slowly decaying pool (100 yr$^{-1}$), 30–40% goes into a medium decay pool (10 yr$^{-1}$), and 60% goes into a rapidly decaying pool (1 yr$^{-1}$). If crops or pasture replace grass, then 100% of the litter goes into the fast decay pool[51].

Litter from natural and pasture PFTs is added to the soil. A fraction of litter from crop PFTs becomes a harvest flux ($H$), which is added to the rapidly decaying

**Table 2 Evaluation of carbon stored in vegetation ($C_{veg}$) and turnover times (defined as soil carbon, $C_{soil}$, divided by net primary productivity, NPP) in the model in the year 2000**

| | $C_{veg}$ (kgC m$^{-2}$) | | Turnover (yr) ($C_{soil}$/NPP) | |
| --- | --- | --- | --- | --- |
| | Observations | Model | Observations | Model |
| Tropical forests | 11 (2.9–19) | 10 (1.7–15) | 11 (6.0–21) | 6 (5.0–10) |
| Mixed forests | 2.9 (0.7–5.8) | 3.9 (0.3–8.3) | 23 (10–43) | 24 (11–47) |
| Boreal forests | 2.0 (0.9–2.9) | 4.6 (0.3–7.6) | 69 (27–147) | 53 (31–157) |
| Tropical savannas | 5.1 (0.2–12) | 4.4 (0–11) | 16 (8.0–81) | 10 (7.0–17) |
| Temperate grasslands | 1.5 (0.3–2.9) | 1.4 (0–4.3) | 38 (12–89) | 45 (17–96) |
| Tundra | 0.5 (0.1–1.0) | 1.5 (0–5.6) | 133 (25–475) | 87 (48–1402) |
| Mediterranean woodlands | 2.4 (0.7–3.7) | 1.5 (0–4.3) | 20 (11–46) | 26 (19–37) |
| Deserts | 0.8 (0.1–2.6) | 0.3 (0–0.4) | 50 (14–174) | 28 (12–57) |

Values are average across 8 biomes based on the World Wildlife Funds 14 eco-regions. The numbers in parentheses are the 10th and 90th percentiles of values in the biome and indicate the scale of spatial heterogeneity

wood product pool and is therefore quickly respired back into the atmosphere:

$$H = \varepsilon_{harv} \Lambda$$
$$\Lambda = \Lambda - H_i$$

Here $\varepsilon_{harv}$ is an efficiency of harvesting and is 0.3, and $H_i$ is the initial harvest flux. If a portion of the cropland includes bioenergy crops, that proportion of the harvest flux is diverted to a BECCS pool instead of being added to the fast wood products pool:

$$BECCS = H_i * \frac{\nu_{bio}}{\nu_{agric}} \varepsilon_{BECCS}$$

and

$$H = H_i - BECCS$$

$\varepsilon_{BECCS}$ is the efficiency term assumed to be 0.6, and $\nu_{bio}$ and $\nu_{agric}$ are the fractions of bioenergy and total crops in the grid box, respectively. This accounts for losses in carbon from harvest to storage in geologic repositories. Unlike the wood products pools, the BECCS pool does not decay, hence captured carbon is kept from the atmosphere indefinitely.

JULES simulations were performed on the N48 grid used in IMOGEN (2.5° latitude × 3.75° longitude). This course resolution allows for many simulations to be run efficiently (each experiment was run with 34 separate climates for the 21st century, resulting in 170 century-long simulations. We regridded the IMAGE data using Patch recovery in the National Center for Atmospheric Research Command Language (NCL) Earth System Modelling Framework regridding toolkit[52]. Land-use data from IMAGE was provided at a 0.5° × 0.5° resolution as fraction of crop, pasture, and bioenergy crop in each grid cell and regridded to N48 using NCL. The regridding results in slightly more land area in the N48 resolution: 13345.8 Mha compared to 12991.1 Mha in the original resolution. The potential impacts on production simulated in JULES are small: land area for bioenergy crops in 2100 is 3% lower after regridding in IM1.9 and 2% higher in IM2.6

The ability of JULES to simulate natural vegetation and soil carbon dynamics has been previously validated[30,53]. We compare simulated and observed biome-average $C_{veg}$ and turnover time (defined as NPP/$C_{soil}$), following the method for calculating biome averages in Harper et al.[23] and after regridding from their native resolutions to the N48 resolution. The relevant observations are above/below-ground biomass carbon[54], NPP from MODIS[55,56], and a global dataset of soil carbon[57] (Table 2). Simulated global vegetation and soil carbon are 510 GtC and 1800 GtC (in the top 1 m) in 2010, respectively, compared to observation-based estimates of 490 GtC and > 2400 GtC. Vegetation carbon and turnover times (calculated as $C_{soil}$/NPP) are similar to observations in most biomes (Table 2). $C_{veg}$ and $C_{soil}$ are too high in boreal forests and tundra. The latter is not affected by land-use change and therefore does not impact the comparison of land-use scenarios in this study. On average, vegetation carbon is overestimated by 2.5 kgC m$^{-2}$ in boreal forests. The maximum area deforested from boreal forests in 2060 in IM1.9 is ~100 Mha, or $10^6$ km$^2$, indicating a possible overestimation of vegetation carbon losses of in JULES on the order of 2.5 GtC. This is within the range of uncertainty of vegetation carbon changes resulting from the spread in future climates (shading in Fig. 2a).

We compare JULES NPP and harvest to the IMAGE non-woody biomass for bioenergy. To convert from kgC NPP to $T_{DM}$ we assume 50% of biomass is carbon, a moisture content of 10%, and 50% of NPP is allocated to aboveground biomass[58]. Figure 7 shows zonal means of the IMAGE yield compared to JULES above-ground NPP and harvest.

**Dynamic global vegetation models**. Our results could be sensitive to model structure and parameterizations, therefore we ran similar simulations with five other DGVMs: ISAM, JSBACH, LPJ-wsl, LPJ-GUESS, and ORCHIDEE-MICT. The simulations used a subset of climates (HadGEM2-ES, CSIRO-Mk-3-6-0, and GFDL-ESM2G) for the 1.5 °C_IM1.9 and 1.5 °C_IM2.6 scenarios. Similar to JULES, these models couple dynamic vegetation with biogeochemical processes to estimate carbon and water fluxes as well as carbon stocks. Vegetation cover is prescribed in JSBACH, while for the other models the natural PFTs compete with each other for light, water, and nutrients and their coverage is predicted by the models. LPJ-wsl, LPJ-GUESS, and ORCHIDEE-MICT were run at the same resolution as JULES, JSBACH was run at T63 resolution (1.9°), and ISAM was run at 0.5° resolution. ISAM, LPJ-GUESS, and JSBACH include an interactive N cycle which can limit vegetation growth.

ISAM[59,60] represents vegetation C and N in 4 pools (leaves, above-ground wood, coarse, and fine roots). In mixed biomes, C and N pools are duplicated for overstory forest and understory grasses. There are four litter pools and four soil organic material pools for both C and N[60], and two inorganic N reservoirs. Deforestation and crop harvest are represented[61]. During deforestation, part of the aboveground carbon in the deforested area is removed and added to the product pools. The rest of the aboveground carbon is treated as litter fall and added to the corresponding soil C pools, effectively representing plant material left on the ground following deforestation activities. The planting date and harvest date for crops are determined using a phenology model[62]. After the crop harvest, part of the crop carbon is transferred into the land-use product pools and part of the carbon is treated as litter input. For the pastureland, no management is considered.

JSBACH[63] represents carbon in green (living parts of plant), reserve (sugars and starches), and wood pools. Dead biomass is transferred to soils via litter and decomposed in the soil carbon model YASSO[64]. There are eight PFTs representing natural vegetation[65], plus C3 and C4 pasture and crop types. Respiration, fire (depending on fuel availability and humidity), and windthrow remove carbon from soils and vegetation. An interactive N cycle includes denitrification, leaching, and deposition (constant after 2014). Extratropical crops are harvested based on accumulated heat during the growth phase, and a second growth phase may follow, while winter crops enter the rest phase. Tropical crops experience random harvest events throughout the year. Harvested carbon is released to the atmosphere within a year. Deforestation results in allocation of most woody biomass to fast (~decades lifespan) and slow (100-year lifespan) product pools, while carbon in leaves, reserve, and the remaining wood pools are directly released to the atmosphere by deforestation fires. Belowground biomass goes to the litter pool.

LPJ-GUESS[66] represents carbon in vegetation (leaves, sapwood, heartwood, roots), soil (eleven pools), and wood products. Competition occurs between different age classes of ten woody PFTs and a herbaceous understory. Disturbances include fires and stochastic patch-destroying events with an expected return interval of 100 years. There is an interactive nitrogen (N) cycle, including N fertilization inputs that were obtained from IMAGE. Pastures are represented as C3 or C4 grasses, where 50% of aboveground biomass is respired each year to represent grazing. Crops are represented with four crop PFTs, with variable sowing and harvest date, harvest (90% of grain and 75% of other aboveground biomass oxidized), tillage, irrigation, N fertilization, and a dynamic potential heat unit calculation. Bioenergy crops were grown as the maize crop PFT (90% of grain and 90% of other aboveground biomass harvested and used for bioenergy with carbon capture and storage). Deforestation results in 74% of woody biomass and 71% of leaves decaying the same year, 20% of woody biomass goes into the product pool (with a 25-year turnover time), and the remainder goes into litter.

LPJ-wsl[67,68] represents carbon in vegetation (leaf, sapwood, heartwood, roots) and soil (fast and slow pools) for nine PFTs. Fire follows a semi-empirical approach relating the probability of daily fire to area burned. Land-cover and land-use change includes deforestation and regrowth, with crops represented by a generic 'pasture' tile. Crop harvest occurs when leaf area index reaches its maximum

potential. Deforestation results in instantaneous emissions accounting for half of the sapwood and heartwood, and all of the leaf and root biomass go to the above and belowground litter pools.

ORCHIDEE-MICT (v8 4.1) has an improved description of high-latitude hydrology and interactions between soil carbon and soil hydrology and thermal processes[69]. There are eight vegetation pools (foliage, above- and below-ground sapwood and heartwood, fruits, roots and carbon reserves) with distinct mortality and turnover rates, and three soil pools (active, slow, and passive). There are eight forest PFTs, C3 and C4 crops, and C3 and C4 natural grasses. Fires are simulated by the process-based prognostic SPITFIRE module[70,71]. When harvested, 45% of crop NPP is removed and consumed directly. Deforestation results in harvesting of carbon in aboveground sap and heartwood. The other biomass pools are transferred to litter, and litter and soil carbon pools are diluted into the litter and soil pools of the increasing PFTs.

**Modelling future climate change**. Earth System Models (ESMs) are the main tool for predicting climate change. Such models are designed to represent the key physical and biogeochemical cycles of the climate system, and how these interact with anthropogenic forcing through fossil fuel burning. ESMs are usually driven by prescribed emissions or atmosphere multi-gas compositions, including greenhouse gases (GHG). However as ESMs have different sensitivities to adjusted GHG composition, the modelled time evolution of warming will be different for the same profile of such gases. To sample this uncertainty in an integrated modelling framework we make use of the Integrated Model Of Global Effects of climatic aNomalies (IMOGEN[27]) climate analog model. IMOGEN combines a global energy balance component along with regional and monthly meteorological changes from "pattern-scaling"[72], all coupled to the JULES terrestrial carbon cycle model. IMOGEN includes a simple representation of the ocean carbon cycle[73]. The IMOGEN EBM and patterns allow interpolation of ESMs to different future scenarios[74]. The IMOGEN system used here is the current version, recently calibrated against 34 ESMs in the CMIP5 ensemble[35]. The pattern-scaling gives changes in the seven surface meteorological quantities needed to drive JULES.

In standard configuration IMOGEN is driven with prescribed atmospheric $CO_2$ concentration and non-$CO_2$ radiative forcings. In instances where IMOGEN is used to test new surface processes, these concentrations are often standard Representative Concentration Pathways (RCPs) derived from[73]. For this study, though, we have inverted the IMOGEN energy balance model, deriving an evolving radiative forcing in time (specific to each ESM emulated) that corresponds to our prescribed global temperature profiles[35]. The result is 34 unique atmospheric $CO_2$ concentrations consistent with the global mean temperature target based on temperature profiles derived from ref.[28]. IMOGEN also provides an estimate of atmosphere-ocean $CO_2$ fluxes from its oceanic response function, while JULES predicts atmosphere-land $CO_2$ fluxes. The sum of these two fluxes plus the change in atmospheric carbon are equal to the "allowable" emissions for stabilization of the climate at the chosen level of global warming. We can therefore diagnose the fossil emissions which are consistent with a given temperature target and mitigation scenario.

The primary scenarios are: a temperature change of 1.5 °C by 2100 with land-use change from IM1.9 (1.5 °C_IM1.9), and a temperature change of 2 °C by 2100 with land-use change from IM2.6 (2 °C_IM2.6). We compare the impacts of land-use and climate change by running the 1.5 °C climate with IM2.6 land use (1.5 °C_IM2.6), and the 2 °C climate with IM1.9 land use (2 °C_IM1.9). A final scenario is a variant of the 2 °C scenario: with 2 °C warming, IM2.6 land-use, but $CO_2$ from the 1.5 °C scenario, to assess the impact of $CO_2$ fertilization on the results.

**Attributing differences in the land carbon between scenarios**. In Fig. 5, the net difference in land carbon is the result of $CO_2$ fertilisation + climate change + LUC for BECCS + LUC for food + LUC for afforestation/reforestation. We diagnose the net difference in land stocks between IM1.9 land use at 1.5 °C warming (1.5 °C_IM1.9) and IM2.6 at 2 °C warming (2 °C_IM2.6). The $CO_2$ effect is isolated by comparing two 2 °C_IM1.9 scenarios: one with the transient $CO_2$ associated with 2 °C warming and one with transient $CO_2$ associated with 1.5 °C warming. The climate change effect is isolated by comparing two IM1.9 land use scenarios with the same $CO_2$ concentrations: 1.5 °C warming with the associated $CO_2$, and 2 °C warming. The LUC effect is based on the difference between IM2.6 and IM1.9 land use both at 1.5 °C warming.

The land-use related differences are further compared in Fig. 3b. LUC for BECCS (or for food) is the change in $C_{veg} + C_{soil}$ on land with bioenergy crops (or with food crops and pasture). Afforestation/reforestation refers to the change in $C_{veg} + C_{soil}$ on land with reduced agriculture between 2000 and 2100. For each grid cell and each year, if agricultural area increases, LUC driven changes in carbon stocks are attributed to BECCS or to food production (comprising both food crops and pasture) by dividing the change in carbon by the relative increases in area of BECCS crops, food crops and pasture. In the years following this land-use change, ongoing changes in carbon stocks are accumulated based on this categorization until a different type of land-use change occurs. Conversely, if agricultural area decreases, changes in carbon are attributed to reforestation.

When calculating pay back times, the change in carbon stocks on land with LUC for bioenergy crops is calculated in any grid cell where bioenergy crops are present, from the year they first occur in the grid cell until either the year bioenergy crops are removed or the end of the simulation (2100).

**Code availability**. JULES is an open-source model and the branch used in this work is available from the Met Office Science Repository using the following URL (registration required): https://code.metoffice.gov.uk/trac/jules/browser/main/branches/dev/edwardcomynplatt/vn4.8_1P5_DEGREES?rev=11764.

**Data Availability**. The data that support the findings of this study are available from the corresponding author upon request. IMAGE scenario land-use data is also available from https://data.knmi.nl/datasets?q=PBL.

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

## Acknowledgements

The work was undertaken as part of the UK Natural Environment Research Council's programme "Understanding the Pathways to and Impacts of a 1.5 °C Rise in Global Temperature" through grants NE/P014941/1 (P.C., A.H., T.P., J.H., C.H., S.Sitch, T.L.), NE/P015050/1 (E.C.-P., G.H., S.C.), and NE/P014909/1 (W.C., C.W., C.H., P.C., S.Sitch). The authors also acknowledge support from: EPSRC Fellowship EP/N030141/1 (A.H.); NERC NE/P019951/1 (A.H. and T.L.); the Joint UK BEIS/Defra Met Office Hadley Centre Climate Programme (GA01101) (E.B., A.W., and E.R.); CRESCENDO (EU project 641816) (E.B., A.W., E.R., and L.B.); EU FP7 LUC4C program (GA603542) (J.H., A.K., S.Sitch, and A.W); CEH National Capability Fund (C.H.); the U.S. National Science Foundation (NSF) under the EPSCoR Track II cooperative agreement no. OIA-1632810 (B.P.); and through NSF-AGS-12-43071 (A.K.J. and S.Shu).

## Author contributions

P.C., A.H., S.S., J.H., and T.L. designed the overall project. A.H. and T.P. led the analysis, but all authors contributed to the interpretation of the results and to writing of the manuscript. C.H. provided IMOGEN calibrated against the CMIP5 archive, and C.H. and E.C.-P. led development of the inverse IMOGEN model. D.V., V.D., and J.D. contributed IMAGE scenario data and expertize on IMAGE. E.R. and A.W. provided expertize on representing land-use change in JULES. E.B. and S.C. contributed expertize on soil carbon modelling in JULES. G.H., E.C.-P., C.H., E.B., S.C., W.C., C.W., P.C., A.H., and T.P. contributed to the design of the IMOGEN model runs. Additional DGVM simulations were provided by: A.B., P.Ci., and N.D. (ORCHIDEE-MICT); L.B. (JSBACH), A.K.J. and S.Sh. (ISAM), A.K. (LPJ-GUESS), and B.P. (LPJ-wsl).

## Additional information

**Competing interests:** The authors declare no competing interests.

