## [Peer Review File · Nature Communications]

Reviewers' comments:

Reviewer #1 (Remarks to the Author):

Nature Communications Review

NCOMMS-17-31906-T

Relative effectiveness of land-based mitigation strategies in stabilising climate change at 1.5°C

Harper, A. et al.

Summary

In this study, the authors investigate the land-climate-carbon cycle interactions of two land-use change scenarios from the IMAGE model (IM1.9 and IM2.6) that correspond to a radiative forcing of 1.9 and 2.6 Wm⁻² by 2100. These spatial patterns of land use are used in the land surface model JULES coupled to the IMOGEN climate analogue model under two future climate scenarios (an idealised asymptote that reaches 1.5°C or 2°C in 2100). The main result is that the land use scenario from IMAGE for SSP2-RCP1.9, which has a greater land area for bioenergy crop production than the SSP2-RCP2.6 (difference of 225 Mha at maximum extent), leads to significant carbon emissions from the soil carbon pool due to land use change. This has a significant impact on the effectiveness of BECCS, i.e. the amount of carbon removed by BECCS, in this study compared to the amount of carbon removed by BECCS in the IMAGE SSP2-RCP1.9 scenario.

This work makes a timely and crucially important contribution to the discussions around 1.5°C and the use of land-based mitigation. My main concern is the lack of detail and quantification in the explanation of the results regarding the difference between this study's carbon removed by BECCS and that in the IMAGE scenario. Overall I find the paper is too short to be understood by a broad readership (I make the word count about 1250 without methods, whilst the brief guide to manuscript submission states an allowable word count of 5000). I also find it lacks reference to the work of others, specifically in terms of placing the results from this study in the context of other work. Given the policy relevance of this work and its important and novel contribution, I think this should be rectified. The paper would benefit from expanding the text, to explain more clearly the work conducted and situate the results in the wider literature. I appreciate that very little work has been conducted on 1.5°C scenarios, but there is work on afforestation, avoided deforestation and large-scale bioenergy use (e.g. Sonntag et al 2017 [doi: 10.1002/2016GL068824] and references therein; Boysen et al, 2017).

1. What are the major claims of the paper?

The major claims of the paper are that the land use change in SSP2-RCP1.9 leads to a significant loss of soil due to land use change. This study provides a quantitative analysis of the concern in the wider literature about careful selection of land for bioenergy crop production due to soil carbon loss from land use change and offers a quantitative limit on global bioenergy crop production (using the assumptions in this study). A further claim is that afforestation and avoided deforestation are in the majority of cases in this study, a better option for carbon removal than BECCS. The latter result seems contingent upon the way BECCS has been implemented in the study.

2. Are they novel and will they be of interest to others in the field? If the conclusions are not original, it would be helpful if you could provide relevant references.

The conclusions are novel, will be of interest to others in the field and beyond, and make an important contribution to the discourse on the use of land-based mitigation for 1.5°C.

3. Is the work convincing, and if not, what further evidence would be required to strengthen the conclusions?

The work is convincing. Appropriate models (land surface model with dynamic vegetation and a climate analogue model) have been used to simulate the land-climate-carbon cycle interactions and impacts of large-scale expansion of bioenergy crop production.

One important result, that I think requires further explanation is the reasons for the difference between this study's 'CCS pool', i.e. the carbon removed via BECCS, and the CO₂ emissions stored from BECCS in IMAGE (30 ±1 GtC compared to 230 GtC by 2100 for the 1.5°C scenario).

4. On a more subjective note, do you feel that the paper will influence thinking in the field? I think the paper will influence thinking in the field. The impact of soil carbon losses through land use change on the efficacy of carbon removal through BECCS is understood to be an important concern (see e.g. Farjday & Mac Dowell, 2017 [Energy & Environmental Science 10:1389]; Vaughan & Gough 2016 [Environmental Research Letters 11:095003]; Kemper 2015 [International Journal of Greenhouse Gas Control 40:401-430]; van Vuuren et al 2010 [Energy Economics 32:1105-1120]). This paper is the first, to my knowledge, to quantify this issue using these type of models.

General Issues

1. Throughout, remove the term 'CCS' at all occurrences and replace with BECCS. CCS refers to a set of technologies that can be amended to energy conversion processes including those using fossil fuels. BECCS and/or carbon removed via BECCS are less prone to misunderstanding and will be understood clearly by a wide readership.

2. It is my understanding that the authors have used the IMAGE spatial land use maps for bioenergy crop production and equated this directly to BECCS. If I am not mistaken, within IMAGE bioenergy crops are used elsewhere in the energy system (i.e. for more than just BECCS) and BECCS is applied to residue feedstocks as well as bioenergy crops. The difference between what the authors have done and what is assumed in IMAGE should be made clearer throughout the manuscript.

3. The carbon removed by BECCS result from the study is 7.6 times smaller than the IMAGE model (using the same land use maps) (lines 108-110). In my opinion, this result could be heavily cited and therefore seems important to understand and explain, yet I find the explanation lacking in detail. The result is explored further in Figure 4(f) where the authors plot multiples of the study's 'CCS pool' against the relative success of BECCS vs forest (%). This includes a quantification of the impact of the IMAGE assumption of using 75% of the aboveground biomass for BECCS (the plus sign markers in the figure).

I think there may be up to four further factors that might explain the result:

(1) The difference in how the authors have defined the 'CCS flux' (60% of harvested biomass, or 18% of crop PFT litter, is geologically stored) compared to the equivalent representation in IMAGE, e.g. what is the equivalent percentage in IMAGE?

(2) The use in IMAGE of a larger biomass resource than just bioenergy crops (up to 50% can come from agricultural and forestry residues – van Vuuren et al 2013 [Clim. Change 118:15–27], see p24, Azar et al 2010 [Clim. Change 100:195-202], see p200; Daioglou, et al 2016 [doi: 10.1111/gcbb.12285] examine in detail residue use in IMAGE). Note Smith et al 2016 appear to

include residues in their estimates (see SM, Table S2) whilst Boysen et al (2017) do not include the use of residues in their estimates.

(3) Possibly counteracting (2) to some extent is that, as I understand it, not all bioenergy is used for BECCS in IMAGE. Quantifying points (2) and (3) would be useful information for the reader.

(4) The difference between the yields assumed in IMAGE compared to the NPP values for C3 & C4 grasses used to represent crops in JULES. At present the authors do not offer any quantitative information on this point, only qualitatively referring to higher yields and the multiplication of their BECCS flux in Figure 4(f).

(5) How much of this difference arises from information lost due to the different spatial resolutions, e.g. land use maps from IMAGE (5 x 5 arcminutes http://themasites.pbl.nl/models/image/index.php/IMAGE_framework/IMAGE_3.0_in_a_nutshell) compared to JULES (e.g. 1.875° x 1.25° in Harper et al 2016, GMD 9:2415-2440, p2424). For example, are there areas in IMAGE that are designated agricultural land, that at the lower resolution in JULES are designated forest?

It is important to address these concerns, as it affects all of the following results that use the cumulated C stored via BECCS: Fig. 2(c) & (d), Fig. 3, Fig. 4 (c)-(f)).

Line by line points

Line 21. I suggest you change 'bioenergy with...' in this line, to 'biomass energy with...' as the latter is what you use in the main text (e.g. line 46) and accurately reflects the BECCS abbreviation.

Line 24. I suggest using carbon dioxide removal rather than negative emissions in this line, as you have not yet introduced negative emissions as a concept (line 42). You could put negative emissions as a keyword.

Line 30. Add to assumptions list the use of agricultural and forestry residues.

Line 32. The 'easily' depends on how you have calculated the carbon stored via BECCS (see concerns above) - perhaps remove easily.

Line 34. Replace AR with afforestation (does not change word count and is easier to read).

Line 43-44. Given the previous sentence, this sentence implies negative emissions are not needed for 2°C target, which as you go on to point out in lines 47-49 they are. I suggest you add, 'as well as negative emissions'.

Line 47. I don't think 'excess' is needed.

Line 49, 64, 77. The abbreviation 'IAM' is introduced in line 49, again in line 77, but only used in isolation once in line 64. It would improve the readability to avoid unnecessary acronyms (there are many acronyms in the manuscript, e.g. you could replace AR and AD with afforestation and avoided deforestation, again for readability).

Line 71. I suggest '....potential carbon stored via BECCS based on permanently storing 60% of carbon from harvested....', is a little more informative.

Line 86. Add 'and the use of agricultural and forestry residues as a biomass energy resource'.

Line 89. As I understand it, not all the biomass resource (from residues and bioenergy crops) is used in the model with CCS. Some biomass energy is used in the energy system without CCS. The way the manuscript is currently written it implies all the bioenergy crops, e.g. Figure 1 (e) & (f), is used for BECCS. If my understanding is correct, then I suggest removing 'with CCS' from line 89.

Line 97. I suggest changing 'captured' to 'stored'.

Line 98. Is the twice as strong land carbon sink result to be expected? Can you explain to the reader why this is the case. How does this compare to other studies?

Line 102: 'land stocks and CCS' could be improved to 'land stocks and geological storage'.

Line 122. BECCS flux

Line 109. I would avoid using the phrase 'BECCS negative emissions calculated in IMAGE', as the negative emissions delivered by a BECCS system are less than the amount of CO₂ stored underground (see Smith & Torn 2013 Clim. Change 118:89-103 Fig 2, p96), due to land use change and process emissions (e.g. fertiliser use, transport, energy conversion losses, energy system emissions). I suggest using the phrase CO₂ stored in geological reservoirs via BECCS.

Line 141: This should be changed from CCS flux to BECCS flux.

Line 151. What does 'CCS production' mean? Do you mean potential carbon removal via BECCS?

Line 151. Doubling predicted BECCS could be achieved by using residues as an additional feedstock. Note the efficiency of carbon capture and storage is around 85-95% (flue gas capture), so changes in this would only be a minor factor.

Line 153. What would the effect of this doubling be on some of the other results presented (Fig 2 (d) and Fig 3)?

Line 154. Are these grid boxes in the areas that have been deforested (abstract 'boreal forest soils') to grow the bioenergy crops?

Line 164. The discussion focuses on the difference between the representation of BECCS in this study and that in IMAGE and is notably lacking in quantification or detail about these differences, e.g. what are the yields assumed in IMAGE? (see comments in General issues).

Line 165. Replace 'CCS flux' with BECCS flux or 'Despite the smaller amount of CO₂ stored via BECCS in these simulations than assumed within IMAGE' or equivalent.

Line 169. What is meant by 'efficiency of CCS'? Is this about flue gas capture rates or energy conversion processes? Or is it about the percentage of C in the harvested biomass that is stored underground?

Line 170. Replace bio-energy with bioenergy for consistent spelling within the manuscript.

Line 173. 'biofuels' is not interchangeable with 'bioenergy'. Biofuels tends to refer to liquid biomass fuels (see Chum et al 2011, Chapter 2 Bioenergy. In IPCC Special Report on Renewable Energy Sources and Climate Change Mitigation). I suggest changing this to bioenergy crops.

Line 175. It's not clear to me that the analysis shows forest conservation and afforestation are less uncertain than BECCS. The effectiveness point is clear from the analysis, but its less clear where the uncertainty around for example, afforestation been addressed in the paper.

Line 178. Table 1. I find the paragraph on lines 325-331 a far clearer explanation of the experiments conducted than the way the information is presented in Table 1. An improvement would be to rename 2°C_1.5CO₂ to 2°C_IM26_1.5CO₂. The phrase 'idealized asymptote to... by 2100' could be moved to the Table legend, leaving just the temperature values in the column. The first four rows of the transient atmospheric CO₂ column could be merged together. This reformatting would make the table easier to read in my opinion.

Line 190. Figure 2. I suggest the use of the term 'CCS' here is misleading for a wide readership, I would recommend something like 'Cumulative geological CO₂ stored'.

Line 190. Figure 2. The sub-figure title 'Total land C stock' would more usually be assumed to be C_{veg} + C_{soil}. I suggest something like 'Sum of panels (a) + (b) + (c)' or 'Sum of Vegetation, soil and geological storage'.

Line 198. Figure 3. Please clarify what is meant by total land C stock and amend as above.

Line 198. Figure 3. The y-axis scale in panel (b) could be set the same as panel (a) thus removing the need for the repetition of 'cumulative carbon stored via BECCS' in the legend and making it a little clearer what panel (b) is showing.

Line 198 Figure 3. Cumulative CCS flux appears twice in the key. Also, please change 'CCS flux' to cumulative BECCS carbon stored or equivalent.

Line 210. Change (b,c) to (c, d).

Line 215-216. What is 'natural CCS flux'? This sentence should be rewritten to remove 'CCS flux' and improve clarity.

Line 218. '...harvesting the initial aboveground biomass as in IMAGE'.

Line 254. The spatial resolution of the modelling work undertaken is not provided in the methods. Given the significant difference between the carbon stored by BECCS in this work, compared to the IMAGE result (lines 108-110), it would be useful to understand if any of this difference arising from a loss of information between the IMAGE spatial land use maps and the resolution that the JULES-IMOGEN configuration used.

Line 262. missing a comma after natural.

Line 283. The 'CCS pool' referring to the fraction of harvested biomass that goes to permanent geological storage (i.e. BECCS) is introduced in the methods section as having a value of 0.6. There is no justification or explanation of the choice of this value. Another way of expressing your BECCS flux seems to be 0.18 of the crop PFT litter. For the 0.6 value, as a minimum it would be useful to refer to similar literatures (e.g. see Smith & Torn, 2017 who suggest 47% but seem to assume all the biomass is used rather than a fraction harvested, or Boysen et al 2017, who use 50%). Given the comparison provided on lines 108-110, it would also be good to find out approximately what the equivalent value is in the IMAGE model.

Line 295 Suggest 'Modelling future climate change' or equivalent rather than 'IMOGEN' as a sub-heading, this is more similar to the previous sub-heading and more informative for the reader.

Line 339 Subscript the 2 in CO₂

SM Figure SM4. Typographical error in the figure legend 'relative'

SM Figure SM4. Both panels have the same panel title 'RCP19 total effect of difference in forest distribution'. This implies to the reader both panels show the same thing and does not seem to clearly relate to the description of the two panels in the figure legend, '...stocks in IM1.9 relative to IM2.6 (left) and in IM2.6 relative to IM1.9 (right)'.

SM Figure SM4. It appears that the unit label for the colour bar is misplaced. It is currently partially covering the panel titles, when it should be below the colour bars or in the figure legend as it is for the previous three SM figures.

Reviewer #2 (Remarks to the Author):

Harper et al. assess the impacts of land use strategies taken to limit global warming below 1.5C and 2.0C. They specifically compare a land use scenario with moderate bioenergy and carbon capture and storage (BECCS) with a new more intensive management scenario that includes more BECCS to meet a 1.5C warming threshold. These two land use scenarios were made from the IMAGE integrated assessment model, and the authors evaluate their impacts on the global carbon budget of terrestrial ecosystems using the JULES land surface model forced with IMOGEN climate that matches the expected temperature trajectory for a 1.5 or 2.0C stabilisation. The IMAGE land use scenario for 1.5C stabilisation is new.

The authors show that the extra carbon dioxide in the 2.0C scenario stimulates a larger terrestrial carbon sink in the JULES model, especially in soils. The carbon sink from the extra CO₂ fertilization exceeds losses from the extra warming in the 2.0C scenario.

The extra land use change necessary to ramp up BECCS to achieve a 1.5C target requires in Jules extra land use change and emissions from natural vegetation biomass and soils. This offsets the value of BECCS considerably. In many areas, even if BECCS efficiency is greatly amplified, this technology is not as efficient as keeping the natural vegetation intact and allowing it to respond rising carbon dioxide.

General comments:

The overall scope of the paper is highly technical. On the one hand, I think the results are important because they explore tradeoffs between BECCS and less intensive management practices that will be valuable for the IPCC, IPCC AR6 and potentially policy makers evaluating the best way to manage land to stabilize climate. On the other hand, the complexity of the analysis, with coupling between an IAM, a land surface model (that solely considers carbon fluxes), and the forcing from a reduced complexity atmospheric pattern model makes the analysis highly technical. The study outcomes are very sensitive to parameterizations in the JULES model.

For a general audience like Nature Communications, the authors should take steps to simplify the presentation of the analysis and strengthen the discussion and conclusion sections to discuss the results in a broader context. I didn't feel that the final discussion/conclusions paragraph adequately discussed uncertainties or assessed the implications of the analysis.

One limitation that is concerning is the sole focus on carbon dioxide. For attaining 1.5C and 2.0C targets, critical progress must be made on reducing N₂O and CH₄. For BECCS to operate successfully in the new IMAGE scenario, what would be the N fertilization requirements? These are massive, fast growing tree or grass plantations, right? My guess is that the extra N₂O production from N fertilizers would be equally important to carbon dioxide in considering the full climate footprint of the enterprise. A simple scaling argument/estimate for nitrogen requirements would be helpful. It might strengthen the authors' conclusions. Recognizing the other gases and changes in

land surface biophysics (e.g. albedo) is important as well to communicate to the reader that the authors understand the complexity of the issues related to global land management.

Another possible inconsistency is hydroelectric power generation. How does this extra afforestation and carbon stocks/NPP for the IM2.6 land use scenario increase ET and reduce runoff for hydroelectric power generation?

Finally, it was really difficult for this reviewer to believe that the soil carbon stocks in JULES would increase in magnitude to store between 5-10 years of contemporary fossil fuel emissions by 2100 (Fig 2b). This is a massive flux for a low to moderate atmospheric CO₂ scenario. Recent work has suggested that CMIP5 models considerably underestimate the age of soil carbon, and therefore overestimate its potential to take up carbon in response to short-term (decadal-scale) NPP increases (He et al., 2016, Science). Are aboveground and belowground carbon residence times and stocks in the version of JULES used for this simulation consistent with available stock and isotopic constraints? The reader needs more confidence that the model simulation of soil carbon is believable.

Specific comments:

Figure 1. In panels e and f, aren't the differences shown 2060-2000 and 2085-2000 (not the opposite shown in the legend)?

Table 1. The information content in this table is low, and it is filled with jargon. Making this table more descriptive and accessible by a wider audience would be very helpful.

Reviewer #3 (Remarks to the Author):

1. This is a well-written paper addressing an important and highly topical problem of wide spread interest to both the climate science and policy communities. In addressing the problem, the authors have drawn upon established and well-regard models and modelling approaches which they have applied with due care. The key model assumptions have been acknowledged and the modelling results accurately interpreted. The conclusions are consistent with and well supported by the results. There are however, some model assumptions that warrant further scrutiny and critical review. I therefore recommend the paper for publication subject to a minor revision that takes into account the following points.

2. Line 74 and 274; the parameterisation of the wood products modelling appears to me unrealistic in terms of the total proportion of tree biomass allocated to wood products and the distribution of this to each pool. Typical industry figures paint quite a different picture; see Keith et al. (2015) Under What Circumstances Do Wood Products from Native Forests Benefit Climate Change Mitigation? PlosOne DOI: 10.1371/journal.pone.0139640. (S2 Table D: & S2 Fig B).

3. Line 292; one of the main conclusion from the paper concerns the situation where BECCS involves replacing high-carbon content ecosystems with crops. While it is true that the ability of JULES to simulate vegetation and soil carbon dynamics has been previously validated, the question is to how validly JULES represents the natural carbon carrying capacity of carbon dense ecosystems such as forests?

3. Line 289; The authors need to further justify the statement that the efficiency of the entire carbon capture and storage is assumed to be 0.6 and that that the CCS pool "does not decay", hence "captured carbon is kept from the atmosphere indefinitely." My understanding is that storage of capture carbon in natural underground cavities is not "forever" as these leak; engineers I have discussed this issue with have suggested a 40-year time horizon. Furthermore, the validity

of the approach is being question in toto as per the recent European Academies report (<https://easac.eu/publications/details/easac-net/>).

4. To conclude, I should add that the three comments above serve only to reinforce and not detract from the main conclusion of the paper, i.e. to increase further the relative mitigation value of AR/AD compared to BECC.

We thank the reviewers for their helpful comments. In this document, the responses to reviewers are in blue. I have highlighted in yellow main points that I believe warrant a response (these were not always part of a direct question). *Please note that any reference to line numbers applies to the "clean" document with new text in blue* (the line numbers are incorrect in the original word document with track changes due to the multiple changes to the file).

Reviewers' comments:

Reviewer #1 (Remarks to the Author):

Nature Communications Review

NCOMMS-17-31906-T

Relative effectiveness of land-based mitigation strategies in stabilising climate change at 1.5°C

Harper, A. et al.

Summary

In this study, the authors investigate the land-climate-carbon cycle interactions of two land-use change scenarios from the IMAGE model (IM1.9 and IM2.6) that correspond to a radiative forcing of 1.9 and 2.6 Wm⁻² by 2100. These spatial patterns of land use are used in the land surface model JULES coupled to the IMOGEN climate analogue model under two future climate scenarios (an idealised asymptote that reaches 1.5°C or 2°C in 2100). The main result is that the land use scenario from IMAGE for SSP2-RCP1.9, which has a greater land area for bioenergy crop production than the SSP2-RCP2.6 (difference of 225 Mha at maximum extent), leads to significant carbon emissions from the soil carbon pool due to land use change. This has a significant impact on the effectiveness of BECCS, i.e. the amount of carbon removed by BECCS, in this study compared to the amount of carbon removed by BECCS in the IMAGE SSP2-RCP1.9 scenario.

This work makes a timely and crucially important contribution to the discussions around 1.5°C and the use of land-based mitigation. My main concern is the lack of detail and quantification in the explanation of the results regarding the difference between this study's carbon removed by BECCS and that in the IMAGE scenario. Overall I find the paper is too short to be understood by a broad readership (I make the word count about 1250 without methods, whilst the brief guide to manuscript submission states an allowable word count of 5000). I also find it lacks reference to the work of others, specifically in terms of placing the results from this study in the context of other work. Given the policy relevance of this work and its important and novel contribution, I think this should be rectified. The paper would benefit from expanding the text, to explain more clearly the work conducted and situate the results in the wider literature. I appreciate that very little work has been conducted on 1.5°C scenarios, but there is work afforestation, avoided deforestation and large-scale bioenergy use (e.g. Sonntag et al 2017 [doi: 10.1002/2016GL068824] and references therein; Boysen et al, 2017).

Thank you for the thorough and helpful comments on the manuscript. Since the original manuscript was prepared for a *Nature* letter format, it was very short. Now that the manuscript is being considered for *Nature Communications*, we have more space to elaborate on the details of the study and to provide more context for the results. We have added more details and context

for the work in the Introduction and Discussion. All changes in the document are tracked with Word 'Track Changes'.

1. What are the major claims of the paper?

The major claims of the paper are that the land use change in SSP2-RCP1.9 leads to a significant loss of soil due to land use change. This study provides a quantitative analysis of the concern in the wider literature about careful selection of land for bioenergy crop production due to soil carbon loss from land use change and offers a quantitative limit on global bioenergy crop production (using the assumptions in this study). A further claim is that afforestation and avoided deforestation are in the majority of cases in this study, a better option for carbon removal than BECCS. The latter result seems contingent upon the way BECCS has been implemented in the study.

2. Are they novel and will they be of interest to others in the field? If the conclusions are not original, it would be helpful if you could provide relevant references.

The conclusions are novel, will be of interest to others in the field and beyond, and make an important contribution to the discourse on the use of land-based mitigation for 1.5°C.

3. Is the work convincing, and if not, what further evidence would be required to strengthen the conclusions?

The work is convincing. Appropriate models (land surface model with dynamic vegetation and a climate analogue model) have been used to simulate the land-climate-carbon cycle interactions and impacts of large-scale expansion of bioenergy crop production.

One important result, that I think requires further explanation is the reasons for the difference between this study's 'CCS pool', i.e. the carbon removed via BECCS, and the CO₂ emissions stored from BECCS in IMAGE (30 ±1 GtC compared to 230 GtC by 2100 for the 1.5°C scenario).

As pointed out by the reviewer, about 50% of this discrepancy is because dedicated bioenergy crops only contribute roughly half of the BECCS in IMAGE (the remainder being made up of agriculture and forestry residues), and the JULES simulations do not account for residues. It is more relevant to compare the JULES BECCS to 130 GtC for the 1.5°C scenario and this is now updated in the main text. (see lines 133-135, 161, 266)

4. On a more subjective note, do you feel that the paper will influence thinking in the field?

I think the paper will influence thinking in the field. The impact of soil carbon losses through land use change on the efficacy of carbon removal through BECCS is understood to be an important concern (see e.g. Farjday & Mac Dowell, 2017 [Energy & Environmental Science 10:1389]; Vaughan & Gough 2016 [Environmental Research Letters 11:095003]; Kemper 2015 [International Journal of Greenhouse Gas Control 40:401-430]; van Vuuren et al 2010 [Energy Economics 32:1105-1120]). This paper is the first, to my knowledge, to quantify this issue using these type of models.

General Issues

1. Throughout, remove the term 'CCS' at all occurrences and replace with BECCS. CCS refers to a

set of technologies that can be amended to energy conversion processes including those using fossil fuels. BECCS and/or carbon removed via BECCS are less prone to misunderstanding and will be understood clearly by a wide readership.

We have tried to make this distinction clearer throughout the text.

2. It is my understanding that the authors have used the IMAGE spatial land use maps for bioenergy crop production and equated this directly to BECCS. If I am not mistaken, within IMAGE bioenergy crops are used elsewhere in the energy system (i.e. for more than just BECCS) and BECCS is applied to residue feedstocks as well as bioenergy crops. The difference between what the authors have done and what is assumed in IMAGE should be made clearer throughout the manuscript.

As discussed below, we have updated the discussion to reflect these differences in model assumptions.

3. The carbon removed by BECCS result from the study is 7.6 times smaller than the IMAGE model (using the same land use maps) (lines 108-110). In my opinion, this result could be heavily cited and therefore seems important to understand and explain, yet I find the explanation lacking in detail. The result is explored further in Figure 4(f) where the authors plot multiples of the study's 'CCS pool' against the relative success of BECCS vs forest (%). This includes a quantification of the impact of the IMAGE assumption of using 75% of the aboveground biomass for BECCS (the plus sign markers in the figure).

By excluding BECCS from residues, carbon removed by BECCS in this study is now 4.3 times smaller than from IMAGE (Lines 265). This is still a large difference and we attempt to explain it in the updated discussion (Lines 263-302, also see below for individual responses).

I think there may be up to four further factors that might explain the result:

(1) The difference in how the authors have defined the 'CCS flux' (60% of harvested biomass, or 18% of crop PFT litter, is geologically stored) compared to the equivalent representation in IMAGE, e.g. what is the equivalent percentage in IMAGE?

In response to these points, it would be useful to first explain the entire chain of events and fate of carbon from field to storage for both JULES and IMAGE. We now include a comparison of the assumptions in the BECCS calculations in the Discussion in the manuscript.

(2) The use in IMAGE of a larger biomass resource than just bioenergy crops (up to 50% can come from agricultural and forestry residues – van Vuuren et al 2013 [Clim. Change 118:15–27], see p24, Azar et al 2010 [Clim. Change 100:195-202], see p200; Daioglou, et al 2016 [doi: 10.1111/gcbb.12285] examine in detail residue use in IMAGE). Note Smith et al 2016 appear to include residues in their estimates (see SM, Table S2) whilst Boysen et al (2017) do not include the use of residues in their estimates.

We thank the reviewer for pointing out this important difference. We do not include residues in our calculations from JULES. BECCS in IMAGE comes from both residues and dedicated bioenergy crops. Accounting for this, the relevant amount of BECCS is 130 GtC and 20 GtC for IM1.9 and IM2.6, respectively (not 232 and 103 as reported in the original manuscript). This brings JULES closer to the IMAGE calculated BECCS. This is now corrected in the manuscript.

(3) Possibly counteracting (2) to some extent is that, as I understand it, not all bioenergy is used for BECCS in IMAGE. Quantifying points (2) and (3) would be useful information for the reader.

It is true that not all bioenergy crops are used with CCS in IMAGE, but they are in JULES. Generally, about For the IM1.9 scenarios this is 71% of the bioenergy crops are used with CCS in 2050, 69% in 2070 and 63% in 2100.

(4) The difference between the yields assumed in IMAGE compared to the NPP values for C3 & C4 grasses used to represent crops in JULES. At present the authors do not offer any quantitative information on this point, only qualitatively referring to higher yields and the multiplication of their BECCS flux in Figure 4(f). See answer to point (1) above. We added Figure 7 to compare the JULES and IMAGE yields.

(5) How much of this difference arises from information lost due to the different spatial resolutions, e.g. land use maps from IMAGE (5 x 5 arcminutes http://themasites.pbl.nl/models/image/index.php/IMAGE_framework/IMAGE_3.0_in_a_nutshell) compared to JULES (e.g. 1.875° x 1.25° in Harper et al 2016, GMD 9:2415-2440, p2424). For example, are there areas in IMAGE that are designated agricultural land, that at the lower resolution in JULES are designated forest?

It is important to address these concerns, as it affects all of the following results that use the cumulated C stored via BECCS: Fig. 2(c) & (d), Fig. 3, Fig. 4 (c)-(f)).

(5) IMAGE data was provided at a 0.5° x 0.5° resolution. This was regridded using Patch recovery in the NCL ESMF regridding toolkit. Regridding resulted in slightly more land area in the N48 resolution: 13345.8 Mha compared to 12991.1 Mha in the original resolution. The figure below shows the time series of area covered by food crops, pasture, and bioenergy crops for the two scenarios with the original and regridded data. There are small differences in land area for bioenergy crops: by 2100 IM1.9 shows 430 Mha on the original grid and 420 Mha on the N48 grid. By 2100, IM2.6 shows 239 Mha on the original grid and 246 Mha on the N48 grid. The potential impacts on production simulated in JULES are small: land area for bioenergy crops in 2100 is 3% lower after regridding in IM1.9 and 2% higher in IM2.6. This information is now included in the Methods (Lines 434-444).

Line by line points

Thank you for the comments – below if the Line number is in blue it means we made the associated change. Further details are provided when necessary.

Line 21. I suggest you change ‘bioenergy with...’ in this line, to ‘biomass energy with...’ as the latter is what you use in the main text (e.g. line 46) and accurately reflects the BECCS abbreviation.

Line 24. I suggest using carbon dioxide removal rather than negative emissions in this line, as you have not yet introduced negative emissions as a concept (line 42). You could put negative emissions as a keyword.

Line 30. Add to assumptions list the use of agricultural and forestry residues.

Line 32. The ‘easily’ depends on how you have calculated the carbon stored via BECCS (see concerns above) - perhaps remove easily.

Line 34. Replace AR with afforestation (does not change word count and is easier to read).

Line 43-44. Given the previous sentence, this sentence implies negative emissions are not needed for 2°C target, which as you go on to point out in lines 47-49 they are. I suggest you add, ‘as well as negative emissions’.

Line 47. I don’t think ‘excess’ is needed. **This sentence was reworded and ‘excess’ was removed.**

Line 49, 64, 77. The abbreviation ‘IAM’ is introduced in line 49, again in line 77, but only used in isolation once in line 64. It would improve the readability to avoid unnecessary acronyms (there are many acronyms in the manuscript, e.g. you could replace AR and AD with afforestation and avoided deforestation, again for readability). **We kept the acronym IAM (but removed repetitive definitions) since with new text the term occurs 4 times in the manuscript. We replaced all occurrences of ‘AR’ with ‘afforestation/reforestation’ and ‘AD’ with ‘avoided deforestation.’**

Line 71. I suggest ‘...potential carbon stored via BECCS based on permanently storing 60% of carbon from harvested...’, is a little more informative.

Line 86. Add ‘and the use of agricultural and forestry residues as a biomass energy resource’.

Line 89. As I understand it, not all the biomass resource (from residues and bioenergy crops) is used in the model with CCS. Some biomass energy is used in the energy system without CCS. The way the manuscript is currently written it implies all the bioenergy crops, e.g. Figure 1 (e) & (f), is used for BECCS. If my understanding is correct, then I suggest removing ‘with CCS’ from line 89.

Line 97. I suggest changing ‘captured’ to ‘stored’.

Line 98. Is the twice as strong land carbon sink result to be expected? Can you explain to the reader why this is the case. How does this compare to other studies?

This is due to both the fertilizing effect of CO₂, and the growth of more high latitude vegetation with the warmer climate. These positive impacts on land carbon of the 2C scenario are partially offset by losses of soil carbon due to higher respiration rates at 2C compared to 1.5C. Although the differences between the 1.5 and 2C climate scenarios are interesting, we note the high

uncertainty in the CO₂ fertilization and so focus the second half of the results on comparing land-use change differences only in the 1.5C scenario. Also we have added a comparison to other DGVMs who completed a subset of the JULES simulations to put the JULES results into context. See Lines 145-148, 180-187, and new lines 215-223.

Line 102: 'land stocks and CCS' could be improved to 'land stocks and geological storage'.

Line 122. BECCS flux

Line 109. I would avoid using the phrase 'BECCS negative emissions calculated in IMAGE', as the negative emissions delivered by a BECCS system are less than the amount of CO₂ stored underground (see Smith & Torn 2013 Clim. Change 118:89-103 Fig 2, p96), due to land use change and process emissions (e.g. fertiliser use, transport, energy conversion losses, energy system emissions). I suggest using the phrase CO₂ stored in geological reservoirs via BECCS. This is a good suggestion and we have updated the wording throughout the manuscript.

Line 141: This should be changed from CCS flux to BECCS flux.

Line 151. What does 'CCS production' mean? Do you mean potential carbon removal via BECCS? Yes, we have changed this to: "CO₂ removal via BECCS"

Line 151. Doubling predicted BECCS could be achieved by using residues as an additional feedstock. Note the efficiency of carbon capture and storage is around 85-95% (flue gas capture), so changes in this would only be a minor factor. We added residues as a potential method for increasing CO₂ removal via BECCS, although this does not explain the difference between JULES and IMAGE (in the new manuscript) since we now only compare to the BECCS from dedicated bioenergy crops.

Line 153. What would the effect of this doubling be on some of the other results presented (Fig 2 (d) and Fig 3)? In general, a doubling of the carbon stored via BECCS would increase total land carbon by 30 GtC in IM1.9 and by 20 GtC in IM2.6. This would make the total changes in land carbon similar between scenarios (Fig. 2d), so globally the land-use changes for BECCS begin to pay off. However on a grid cell basis our analysis shows about half of the locations with BECCS would still be better off with forests. Also it depends on how the gains are achieved. We can only increase BECCS by 67% by changing the efficiency term in JULES (from 0.6 to 1.0) – this would simply double the BECCS calculated in JULES. Higher gains would come from increasing crop yields, since some carbon also would be added to soils based on the calculation of harvest yields in JULES. The details of the impacts depend on these factors and would require several extra simulations to calculate, and so we have simply added: "... although globally the carbon stocks in IM1.9 would begin to match those in IM2.6 (Figure 2c, d)." in the new manuscript at Line 242-243.

Line 154. Are these grid boxes in the areas that have been deforested (abstract 'boreal forest soils') to grow the bioenergy crops? Yes, they are the grid cells with very long payback timescales. We have added a sentence to explain this.

Line 164. The discussion focuses on the difference between the representation of BECCS in this study and that in IMAGE and is notably lacking in quantification or detail about these differences, e.g. what are the yields assumed in IMAGE? (see comments in General issues). There is now a much more detailed discussion attempting to quantify the primary differences relevant for this

study.

Line 165. Replace 'CCS flux' with BECCS flux or 'Despite the smaller amount of CO₂ stored via BECCS in these simulations than assumed within IMAGE' or equivalent. **We have removed this part of the sentence.**

Line 169. What is meant by 'efficiency of CCS'? Is this about flue gas capture rates or energy conversion processes? Or is it about the percentage of C in the harvested biomass that is stored underground? **We meant the latter, so 'efficiency of CCS' was replaced with 'the amount of harvested carbon that is ultimately stored underground'**

Line 170. Replace bio-energy with bioenergy for consistent spelling within the manuscript.

Line 173. 'biofuels' is not interchangeable with 'bioenergy'. Biofuels tends to refer to liquid biomass fuels (see Chum et al 2011, Chapter 2 Bioenergy. In IPCC Special Report on Renewable Energy Sources and Climate Change Mitigation). I suggest changing this to bioenergy crops.

Line 175. It's not clear to me that the analysis shows forest conservation and afforestation are less uncertain than BECCS. The effectiveness point is clear from the analysis, but its less clear where the uncertainty around for example, afforestation been addressed in the paper.

True, this does not address uncertainty of forest-based mitigation so we have removed the text 'and less uncertain'.

Line 178. Table 1. I find the paragraph on lines 325-331 a far clearer explanation of the experiments conducted than the way the information is presented in Table 1. An improvement would be to rename 2°C_1.5CO₂ to 2°C_IM26_1.5CO₂. The phrase 'idealized asymptote to... by 2100' could be moved to the Table legend, leaving just the temperature values in the column. The first four rows of the transient atmospheric CO₂ column could be merged together. This reformatting would make the table easier to read in my opinion.

We have reformatted the table to make it clearer.

Line 190. Figure 2. I suggest the use of the term 'CCS' here is misleading for a wide readership, I would recommend something like 'Cumulative geological CO₂ stored'.

Line 190. Figure 2. The sub-figure title 'Total land C stock' would more usually be assumed to be Cveg + Csoil. I suggest something like 'Sum of panels (a) + (b) + (c)' or 'Sum of Vegetation, soil and geological storage'.

Line 198. Figure 3. Please clarify what is meant by total land C stock and amend as above. **We added this information in the caption.**

Line 198. Figure 3. The y-axis scale in panel (b) could be set the same as panel (a) thus removing the need for the repetition of 'cumulative carbon stored via BECCS' in the legend and making it a little clearer what panel (b) is showing.

Line 198 Figure 3. Cumulative CCS flux appears twice in the key. Also, please change 'CCS flux' to cumulative BECCS carbon stored or equivalent.

Line 210. Change (b,c) to (c, d).

Line 215-216. What is 'natural CCS flux'? This sentence should be rewritten to remove 'CCS flux' and improve clarity. We have updated the caption to make it clearer.

Line 218. '...harvesting the initial aboveground biomass as in IMAGE'.

Line 254. The spatial resolution of the modelling work undertaken is not provided in the methods. Given the significant difference between the carbon stored by BECCS in this work, compared to the IMAGE result (lines 108-110), it would be useful to understand if any of this difference arising from a loss of information between the IMAGE spatial land use maps and the resolution that the JULES-IMOGEN configuration used. We have added some explanation in the Methods.

Line 262. missing a comma after natural.

Line 283. The 'CCS pool' referring to the fraction of harvested biomass that goes to permeant geological storage (i.e. BECCS) is introduced in the methods section as having a value of 0.6. There is no justification or explanation of the choice of this value. Another way of expressing your BECCS flux seems to be 0.18 of the crop PFT litter. For the 0.6 value, as a minimum it would be useful to refer to similar literatures (e.g. see Smith & Torn, 2017 who suggest 47% but seem to assume all the biomass is used rather than a fraction harvested, or Boysen et al 2017, who use 50%). Given the comparison provided on lines 108-110, it would also be good to find out approximately what the equivalent value is in the IMAGE model.

The efficiency factor is now introduced in the main text (Lines 100-102) and more justification is given, particularly in the Discussion (Lines 276-281).

Line 295 Suggest 'Modelling future climate change' or equivalent rather than 'IMOGEN' as a sub-heading, this is more similar to the previous sub-heading and more informative for the reader.

Line 339 Subscript the 2 in CO₂

SM Figure SM4. Typographical error in the figure legend 'relative'

With Nature Communications we can have more figures in the main text, so we moved SM Figures 2 and 3 to the main text but removed SM Figure 4.

SM Figure SM4. Both panels have the same panel title 'RCP19 total effect of difference in forest distribution'. This implies to the reader both panels show the same thing and does not seem to clearly relate to the description of the two panels in the figure legend, '....stocks in IM1.9 relative to IM2.6 (left) and in IM2.6 relative to IM1.9 (right)'. We removed SM Figure 4.

SM Figure SM4. It appears that the unit label for the colour bar is misplaced. It is currently partially covering the panel titles, when it should be below the colour bars or in the figure legend as it is for the previous three SM figures. We removed SM Figure 4.

Reviewer #2 (Remarks to the Author):

Harper et al. assess the impacts of land use strategies taken to limit global warming below 1.5C and 2.0C. They specifically compare a land use scenario with moderate bioenergy and carbon capture and storage (BECCS) with a new more intensive management scenario that includes more BECCS to meet a 1.5C warming threshold. These two land use scenarios were made from the IMAGE integrated assessment model, and the authors evaluate their impacts on the global carbon budget of terrestrial ecosystems using the JULES land surface model forced with IMOGEN climate that matches the expected temperature trajectory for a 1.5 or 2.0C stabilisation. The IMAGE land use scenario for 1.5C stabilisation is new.

The authors show that the extra carbon dioxide in the 2.0C scenario stimulates a larger terrestrial carbon sink in the JULES model, especially in soils. The carbon sink from the extra CO₂ fertilization exceeds losses from the extra warming in the 2.0C scenario.

The extra land use change necessary to ramp up BECCS to achieve a 1.5C target requires in Jules extra land use change and emissions from natural vegetation biomass and soils. This offsets the value of BECCS considerably. In many areas, even if BECCS efficiency is greatly amplified, this technology is not as efficient as keeping the natural vegetation intact and allowing it to respond rising carbon dioxide.

General comments:

The overall scope of the paper is highly technical. On the one hand, I think the results are important because they explore tradeoffs between BECCS and less intensive management practices that will be valuable for the IPCC, IPCC AR6 and potentially policy makers evaluating the best way to manage land to stabilize climate. On the other hand, the complexity of the analysis, with coupling between an IAM, a land surface model (that solely considers carbon fluxes), and the forcing from a reduced complexity atmospheric pattern model makes the analysis highly technical. **The study outcomes are very sensitive to parameterizations in the JULES model.**

We evaluated the change in land carbon stocks over the 21st century in 5 other dynamic global vegetation models to check if certain conclusions are model-dependent. All 5 models found more land carbon in IM2.6 than in IM1.9. The difference in ΔC_{veg} between scenarios is significantly larger than in JULES in two models (JSBACH and LPJ), but it is always negative. Change in C_{soil} (including litter) over the 21st century is positive in two of the models. This is now discussed at Lines 180-187.

LPJ-GUESS is the only other model with a representation of bioenergy crops with CCS. Using the same efficiency as in this study, the carbon captured via BECCS in LPJ-GUESS would be 73 GtC in IM1.9 and 57 GtC in IM2.6. These numbers are 2.5-3 times higher than predicted by JULES (and also assume that all bioenergy crops are used with CCS). We add this into the Discussion to help explain the potential for higher levels of carbon captured with BECCS than simulated by JULES (See Lines 293-297)

For a general audience like Nature Communications, the authors should take steps to simplify the presentation of the analysis and strengthen the discussion and conclusion sections to discuss the results in a broader context. I didn't feel that the final discussion/conclusions paragraph adequately discussed uncertainties or assessed the implications of the analysis.

Thank you for the helpful comments on the manuscript. Since the original manuscript was prepared for a *Nature* letter format, it was very short. Now that the manuscript is being

considered for *Nature Communications*, we have more space to elaborate on the details of the study and to provide more context for the results. We have added more details and context for the work throughout the Introduction and Discussion.

One limitation that is concerning is the sole focus on carbon dioxide. For attaining 1.5C and 2.0C targets, critical progress must be made on reducing N₂O and CH₄. For BECCS to operate successfully in the new IMAGE scenario, what would be the N fertilization requirements? These are massive, fast growing tree or grass plantations, right? My guess is that the extra N₂O production from N fertilizers would be equally important to carbon dioxide in considering the full climate footprint of the enterprise. A simple scaling argument/estimate for nitrogen requirements would be helpful. It might strengthen the authors' conclusions. Recognizing the other gases and changes in land surface biophysics (e.g. albedo) is important as well to communicate to the reader that the authors understand the complexity of the issues related to global land management. The Discussion is now expanded to describe the implications of the results in more detail, with a particular focus on emissions of other GHGs, impacts on runoff, and biophysical effects of the LUC. See Lines 303-338.

Another possible inconsistency is hydroelectric power generation. How does this extra afforestation and carbon stocks/NPP for the IM2.6 land use scenario increase ET and reduce runoff for hydroelectric power generation?

In the discussion, we now mention biophysical and water cycle implications of the extra LUC for the 1.5C target. We include a discussion of changes in runoff between simulations (Lines 321-328), although we do not fully explore the impacts on the hydroelectric power generation, since the IMAGE model includes an energy systems model that predicts the supply of hydroelectric power, and this is separate from the land surface model that predicts changes in water availability due to land-use change.

Finally, it was really difficult for this reviewer to believe that the soil carbon stocks in JULES would increase in magnitude to store between 5-10 years of contemporary fossil fuel emissions by 2100 (Fig 2b). This is a massive flux for a low to moderate atmospheric CO₂ scenario. Recent work has suggested that CMIP5 models considerably underestimate the age of soil carbon, and therefore overestimate its potential to take up carbon in response to short-term (decadal-scale) NPP increases (He et al., 2016, Science). Are aboveground and belowground carbon residence times and stocks in the version of JULES used for this simulation consistent with available stock and isotopic constraints? The reader needs more confidence that the model simulation of soil carbon is believable.

The new Figure 3 shows where these increases occur: Nearly a third of the increase occurs in the Tundra biome, due to enhanced growth of woody vegetation in response to higher temperatures and CO₂ and a reduction in land covered by bare soil (Figure 4). See updated Lines 145-146, 180-187. Under present-day climate, turnover times (defined as Cs/NPP) are similar to those derived from WISE soil carbon and MODIS NPP, with the exception of the tundra biome. In the tundra, the turnover is too slow due to underestimated Cs and/or overestimated NPP. We have added Table 2 and the relevant discussion in the Methods (Lines 445-460), as well as pointing the reader to previous studies in which vegetation and soil pools have been evaluated in JULES.

Specific comments:

Figure 1. In panels e and f, aren't the differences shown 2060-2000 and 2085-2000 (not the opposite shown in the legend)? Yes we have made this change.

Table 1. The information content in this table is low, and it is filled with jargon. Making this table more descriptive and accessible by a wider audience would be very helpful. We have updated the table to make it clearer.

Reviewer #3 (Remarks to the Author):

1. This is a well-written paper addressing an important and highly topical problem of wide spread interest to both the climate science and policy communities. In addressing the problem, the authors have drawn upon established and well-regard models and modelling approaches which they have applied with due care. The key model assumptions have been acknowledged and the modelling results accurately interpreted. The conclusions are consistent with and well supported by the results. There are however, **some model assumptions that warrant further scrutiny and critical review.** I therefore recommend the paper for publication subject to a minor revision that takes into account the following points.

We thank the reviewer for the constructive and helpful comments. Our responses are below in blue.

2. Line 74 and 274; **the parameterisation off the wood products modelling appears to me unrealistic in terms of the total proportion of tree biomass allocated to wood products and the distribution of this to each pool.** Typical industry figures paint quite a different picture; see Keith et al. (2015) Under What Circumstances Do Wood Products from Native Forests Benefit Climate Change Mitigation? PlosOne DOI: 10.1371/journal.pone.0139640. (S2 Table D: & S2 Fig B). Thank you for pointing this out – we made an error in the text, and the actual allocation for the fast and slow pools are opposite to what was in the original manuscript. This is now corrected. These allocations come from data in McGuire et al. 2001. The allocation of 60% of woody biomass into the fast pool is consistent with the rapid loss of biomass found in Keith et al. (2015) (65% of harvest left on-site as waste/slash by-products). We also reworded the description of allocation of biomass into woody product pools following deforestation to make it clearer.

3. Line 292; one of the main conclusion from the paper concerns the situation where BECCS involves replacing high-carbon content ecosystems with crops. While it is true that the ability of JULES to simulate vegetation and soil carbon dynamics has been previously validated, the question is to **how validly JULES represents the natural carbon carrying capacity of carbon dense ecosystems such as forests?**

We have added a short evaluation of the soil and vegetation carbon in the Methods, and also refer to several recent model development papers (Lines 445-460).

3. Line 289; **The authors need to further justify the statement that the efficiency of the entire carbon capture and storage is assumed to be 0.6 and that that the CCS pool "does not decay", hence "captured carbon is kept from the atmosphere indefinitely."** My understanding is that storage of capture carbon in natural underground cavities is not "forever" as these leak; engineers I have discussed this issue with have suggested a 40-year time horizon. Furthermore, the validity of the approach is being question in toto as per the recent European Academies report (<https://easac.eu/publications/details/easac-net/>).

We have added some text in the discussion acknowledging the importance of a permanent reservoir, along with some other technical issues facing BECCS. (Lines 344-348)

4. To conclude, I should add that the three comments above serve only to reinforce and not detract from the main conclusion of the paper, i.e. to increase further the relative mitigation value of AR/AD compared to BECC.

REVIEWERS' COMMENTS:

Reviewer #1 (Remarks to the Author):

The authors have addressed all the points raised in the original review, in a thorough and detailed manner.